# Evaluating the Rationale Understanding of Critical Reasoning in Logical Reading Comprehension

**Akira Kawabata**
The Asahi Shimbun Company
kawabata-a@asahi.com

**Saku Sugawara**
National Institute of Informatics
saku@nii.ac.jp

## Abstract

To precisely evaluate a language model's capability for logical reading comprehension, we present a dataset for testing the understanding of the rationale behind critical reasoning. For questions taken from an existing multiple-choice logical reading comprehension dataset, we crowdsource rationale texts that explain why we should select or eliminate answer options, resulting in 3,003 multiple-choice subquestions that are associated with 943 main questions. Experiments on our dataset show that recent large language models (e.g., InstructGPT) struggle to answer the subquestions even if they are able to answer the main questions correctly. We find that the models perform particularly poorly in answering subquestions written for the incorrect options of the main questions, implying that the models have a limited capability for explaining why incorrect alternatives should be eliminated. These results suggest that our dataset encourages further investigation into the critical reasoning ability of language models while focusing on the elimination process of relevant alternatives.

## 1  Introduction

Critical reasoning, a type of logical reasoning not tied to formal logic, is a core ability of humans that is required for thoughtful reading of text. It involves not only understanding what a passage explicitly says but also comprehending its underlying assumptions, argument structure, and supported conclusions. Developing systems capable of critical reasoning as reliably as humans is one of the ultimate goals of natural language processing. Recent studies have proposed datasets that evaluate logical reasoning including critical reasoning ability (Yu et al., 2020; Liu et al., 2020) in reading comprehension. Owing to the recent development of large language models (LLMs; Brown et al., 2020; He et al., 2023), the performance of the state-of-the-art models is nearing that of humans (Jiao et al., 2022; Wang et al., 2022).

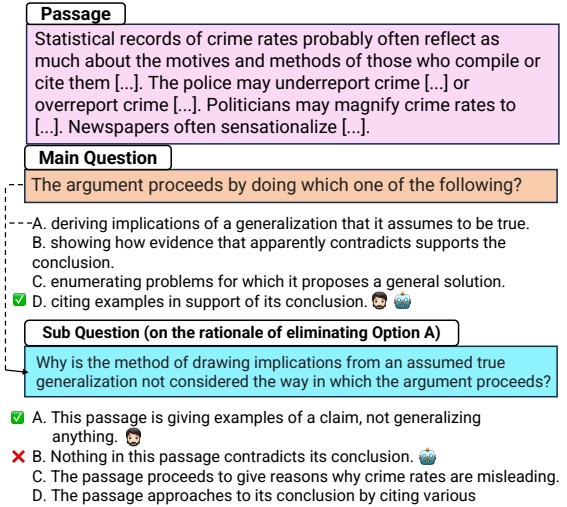

Figure 1: Example of ReClor (Yu et al., 2020) and its subquestion we create to test the understanding of implicit rationale. We find that even if the model can answer the original question correctly, it cannot answer subquestions that should be answerable.

However, current multiple-choice questions in existing logical reading comprehension datasets may not sufficiently test the ability of critical reasoning. The example illustrated in Figure 1 shows that even if a model can answer a question taken from the ReClor dataset (Yu et al., 2020) that has questions for graduate admission examinations, it cannot answer an auxiliary question that queries the implicit rationale for eliminating a relevant alternative. This behavior might be due to the model's limited generalizability that is exposed by input perturbation (Si et al., 2021; Lin et al., 2021; Shi et al., 2023) or characterized as shortcut reasoning (Niven and Kao, 2019; Geirhos et al., 2020). Because a single question cannot fully ask the rationale of why we select an option as the correct answer and eliminate the others as the incorrect ones, current datasets may not be sufficient to comprehensively evaluate the process of critical reasoning.

Recent studies propose methods for probing the

reasoning process using auxiliary generation tasks such as in the form of simple commonsense facts (Aggarwal et al., 2021), logical graphs (Huang et al., 2022), and arithmetic equations (Ribeiro et al., 2023). However, this line of approach may not be suitable to capture the implicit rationale of critical reasoning. In particular, it cannot explicitly consider the selection and elimination process of relevant alternatives in logical reasoning. In addition, the format of such auxiliary tasks is usually not the same as that of the main task, which may fail to evaluate the target abilities consistently.

As a first step to address these limitations, we construct a benchmark that comprehensively evaluates language models' ability of critical reasoning in logical reading comprehension. Our dataset, rationale understanding for logical reasoning evaluation (RULE), consists of main questions taken from ReClor and auxiliary subquestions that we newly create for this study. The process of constructing our dataset is illustrated in Figure 2. Our core idea is that for each answer option in a main question, we crowdsource a free-form human-written rationale that explains why that option should be selected or eliminated, and use those rationales to create a set of subquestions that are associated with the main question. After manual filtering to ensure human answerability, in addition to 943 main questions, we obtain 3,003 subquestions for the test-only purpose. The common multiple-choice format of the main questions and subquestions enables us to evaluate the models' capability of critical reasoning concisely and consistently.

In our experiments using strong baseline models including LLMs, e.g., Flan-UL2, (Tay et al., 2023), LLaMA 2 (Touvron et al., 2023b), and Instruct-GPT (Ouyang et al., 2022), we observe that the models cannot answer the main questions and subquestions consistently, showing a larger than 30% gap against humans in our strict consistency metric. In particular, we find that the models struggle to answer eliminative subquestions, which are pertinent to the rationale of eliminating incorrect options, showing a large gap ($\approx$ 20% accuracy) between humans and the best-performing LLM. Conversely, the models tend to correctly answer selective subquestions, which are pertinent to the rationale of selecting the correct option. This clear contrast suggests that these models provide the correct answer without fully understanding why the other options are incorrect. Our analysis using a follow-up task

and manual annotations supports this observation. We also compare our human-written rationales with model-generated ones using an LLM, finding that our rationales are likely to be more detailed and supportive than the model-generated ones.

Our contributions are as follows: (i) Based on an existing logical reading comprehension dataset, we create a dataset including over 3,000 auxiliary questions designed to test a model's consistent ability for critical reasoning. (ii) We evaluate cutting-edge models, including LLMs, across finetuned, few-shot, and zero-shot settings, showing that even the best model falls short of human performance, particularly lagging in understanding eliminative rationales for incorrect answer options. (iii) Our annotation analysis also highlights the model's deficiency in understanding eliminative rationales and shows that our human-written rationales are of higher quality than model-generated ones.[1]

## 2 Related Works

**Critical and Logical Reasoning**    Critical reasoning is one of the core abilities of logical reasoning that humans perform, along with analytical reasoning (Zhong et al., 2022) and abductive reasoning (Bhagavatula et al., 2020). This reasoning is related to understanding the structure of practical arguments that is generally composed of ground (premise), warrant (rationale), and claim (conclusion). As formulated by Toulmin (2003), given facts or data as the ground, we provide the warrant that acts as a bridge between the ground and the claim we are making. Recent research includes developing ways to model this behavior in tasks such as argument mining and question answering (QA) (e.g., ReClor). For example, Habernal et al. (2018) propose a task of identifying implicit rationale (i.e., warrant) in arguments. However, Niven and Kao (2019) find that successful systems on the argument reasoning task exploit superficial input features. Similarly, QA systems have been shown to exhibit shallow understanding by input perturbation (Si et al., 2021; Lin et al., 2021; Shi et al., 2023). For example, Lin et al. (2021) demonstrate that QA performance significantly decreases when incorrect options are replaced with irrelevant texts in an adversarial manner. This means that successful models on those datasets do not nec-

---

[1]Our dataset, evaluation scripts with model hypterpa-rameters, and annotation results are publicly available at `github.com/nii-cl/rule`

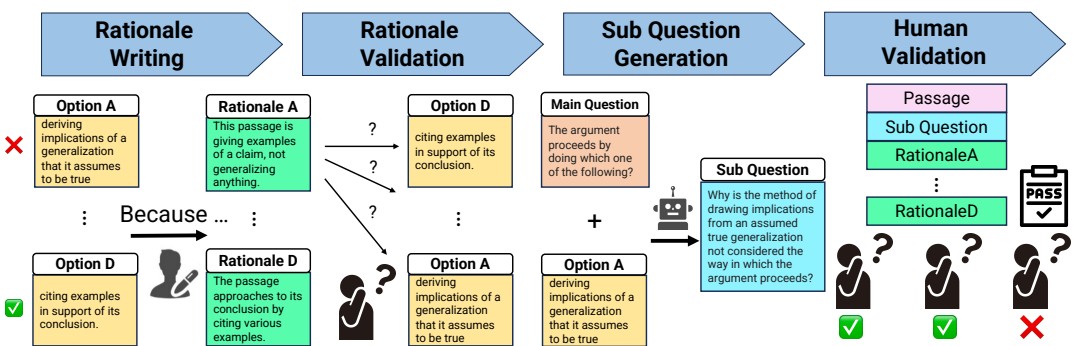

Figure 2: Our dataset construction process. We first ask crowdworkers to write the rationale for each answer option. After validating the collected rationale by aligning them to the source options, we use a large language model to generate subquestion texts. We finally ensure the human answerability of the generated subquestions.

essarily exhibit generalizable capabilities in other datasets. These findings necessitate the explainability of the (informal) logical reasoning process for better evaluation of intended reasoning abilities (e.g., the critical reasoning in this study).

**Reasoning Explanation** Although some studies explain the rationale behind commonsense and logical reasoning using graphs (Saha et al., 2021; Ribeiro et al., 2023), others explain it as a decomposition (Khot et al., 2020; Dalvi et al., 2021; Geva et al., 2021), a combination of supporting textual spans in the input (Yang et al., 2018; Inoue et al., 2020), commonsense rules (Saha et al., 2022), or underlying facts (Aggarwal et al., 2021). The work most similar to ours is MetaLogic (Huang et al., 2022), which focuses on generating graphs explaining the logical relations between sentences in ReClor examples, aiming to model the valid reasoning process. In contrast, we employ free-text rationales that explain the process of critical reasoning, enabling us to construct multiple-choice questions about the understanding of rationales. We also aim to faithfully test the models' performance on the main questions as well as auxiliary subquestions in the multiple-choice discrimination task, instead of the generation of the reasoning process in a different format from the original task.

## 3 RULE Data Collection

### 3.1 Design Choices

We construct a new dataset, RULE (rationale understanding for logical reasoning evaluation), to evaluate the consistent rationale understanding in logical reading comprehension. The dataset comprises main questions and their auxiliary questions (subquestions). The subquestions are designed to test the understanding of the rationale necessary for answering the main questions correctly. In constructing our dataset, we make three decisions in its design choices.

**Source Dataset** Among existing datasets for testing logical reading comprehension, we use ReClor for the following reasons: (1) It covers various types of logical reasoning required in the multiple-choice format, (2) its context passages are of sufficient length to compose a meaningful rationale (e.g., the contexts in LogiQA (Liu et al., 2020) are shorter), and (3) it contains a sufficient number of examples to create an auxiliary benchmarking dataset. We cannot find other candidate datasets, but our approach is applicable to similar ones.

**Rationale Collection** The task of writing implicit rationales from scratch for logical reasoning questions is not straightforward because the reasoning process can involve multiple steps with differing granularity. Therefore, to facilitate rationale writing, we use answer options in the multiple-choice questions. To answer a question with four options, the reasoning process should involve the rationale of both identifying the correct option and eliminating the three incorrect options. By focusing on the correctness of each option, we can decompose the complex task of rationale writing into smaller intuitive tasks. In addition, we collect human-written free-form rationales to expect benefits over model-generated rationales (Sun et al., 2022), in particular for covering the implicit process of critical reasoning.

**Task Format** We also aim to design auxiliary questions so that we can easily evaluate models on both main questions and subquestions in the same task format. To this end, we use four rationales

collected for a main question as the four answer options of its subquestion. A single main question has at most four subquestions that share the same set of answer options, which can be seen as question-wise contrastive evaluation (Gardner et al., 2020; Ashida and Sugawara, 2022).

## 3.2 Collecting Rationales

We use crowdsourcing to collect rationales for creating our subquestions. Appendix A shows our crowdsourcing instructions and examples.

**Qualification** We conduct a two-stage qualification test to recruit crowdworkers for our tasks. The first stage is a QA task to identify workers who carefully answer logical reading comprehension questions. The task consists of ten questions taken from ReClor, and workers achieving $\geq 80\%$ accuracy advance to the next test. In the second stage, workers are presented with a single ReClor question that is randomly sampled from a pool of ten questions. The task is to write four implicit rationales (one sentence each) behind each option's (in)correctness. To guide them, we provide detailed instructions with eight writing examples.

Through preliminary pilot studies, we define two essential criteria for writing rationales: specificity and necessity. Specificity requires rationales to be well informed and support the corresponding options exclusively. This requirement is crucial because non-specific rationales could support multiple options, rendering them unsuitable for options in subquestions. Necessity emphasizes the importance of ensuring that the rationale is essential for validating the option's correctness. Even if a detailed rationale is provided, it must be aligned with the main question's point to preserve its validity.

Following these criteria, the authors manually assess the rationales provided by the workers. We identify 57 workers through this qualification process. These workers are invited to both the rationale writing and subsequent validation tasks.

**Rationale Writing** We take 1,200 questions from the training set of ReClor. As with the second phase of the qualification task, we present workers with a context, question, and four options marked as either correct or incorrect, and then ask workers to write rationale sentences for each option. Of these qualified individuals, 50 were actively engaged in this task. We collect 4,800 rationales in total and send them to the rationale validation step.

**Rationale Validation** To validate the collected rationales, we first focus on their specificity, which is critical for creating a set of reasonable subquestions about a given main question. Because assessing the necessity of rationales may not be straightforward, we analyze the reasoning types involved in understanding rationales in Section 5.

For the validation, we conduct an alignment test between a set of rationales and answer options. In this test, workers are presented with one main question, its four options, and one rationale. They are then asked to identify which one of the options is supported by the given rationale. If a rationale is insufficiently detailed and could potentially support other options, it would be difficult for workers to correctly match the rationale to its corresponding option. We ensure that the worker who validates a rationale is different from the one who wrote it.

This test enables us to refine our initial pool of 4,800 rationales down to 3,828, ensuring that each rationale is sufficiently specific to support its corresponding option.

## 3.3 Subquestion Construction

**Question Generation** We then generate question texts to construct subquestions using a language model. Given one main question and one of its options, the model is instructed to generate a subquestion that asks about the reason for the correctness of the option. For example, when we input the prompt "What mistake does the argument make in its reasoning?" and the incorrect answer option "It confuses probability and certainty," the model generates the question "What evidence is there that the argument does not make the mistake of confusing probability and certainty?" We use different prompts for the correct and incorrect options to avoid the problem of the model omitting negatives (e.g., "not") when generating eliminative subquestions. For the generation, we use Instruct-GPT (`text-davinci-003`), which is one of the strong large language models. Appendix B shows an example of our prompt.

**Subquestion Construction** Coupling the validated rationales with generated question texts, we construct at most four subquestions for a single main question. Each subquestion corresponds to each of the four answer options in the main question. The four answer options of the subquestions are identical to the four rationales written for the main question. The correct answer option of a sub-

question is the rationale written for the option that the subquestion is made from.

A subquestion must have four validated rationales to compose the multiple-choice format. However, when we look at a main question, all four rationales are not always valid, which could largely decrease the number of possible subquestions. To mitigate this issue, we create a subquestion even if three out of the four rationales are valid, by replacing the invalid rationale with the "None of the above choices" option. Through this process, we obtain 3,824 subquestions. We discard a main question if it has no valid subquestions.

### 3.4 Human Validation

As the final step of our data collection, we validate the answerability of the subquestions by humans. Despite the ensured specificity of rationales, the complexity of the subquestion texts could potentially make the subquestions unanswerable. To address this issue, we ask three workers to answer each subquestion to evaluate its human answerability. A subquestion is considered answerable if at least two workers answer it correctly, or if all workers select "None of the above choices." In the latter scenario, we replace the correct answer in the question with "None of the above choices." This process results in 3,003 answerable subquestions with 943 main questions. We expect the number of questions in our dataset can demonstrate statistical power for meaningful model benchmarking and comparison (Card et al., 2020).

We then ask different workers to answer the questions, collecting three additional labels for each question to measure human accuracy.

### 3.5 Dataset Statistics

Table 1 shows the dataset statistics. Compared to the main questions (ReClor), our subquestions have longer questions and answer options. The subquestions that have "None of the above choices" as the correct answer comprise 7.4% (222/3,003) of the dataset, which is comparable to a similar multiple-choice reading comprehension dataset (6.7% in CosmosQA; Huang et al., 2019). We also report the crowdsourcing details in Appendix C.

## 4 Baseline Performance on RULE

We measure the baseline performance of recent state-of-the-art models on our dataset. Because the main purpose of our dataset is to perform an exten-

| # Main / Sub Questions | 943 / 3,003 |
| # SubQ / MainQ | 3.18 |
| # Selective / Eliminative (S/E) | 785 / 2,218 |
| Avg. context length | 73.8 |
| Avg. question length | 31.4 (15.5) |
| Avg. option length | 23.5 (17.7) |
| Avg. correct option length | 24.0 (18.6) |
| # Question vocabulary | 8,843 (1,085) |
| # Option vocabulary | 9,849 (9,652) |
| # SubQ w/ "None" (# answer) | 1,102 (222) |

Table 1: Dataset statistics of our RULE dataset. *S/E* indicates the numbers of two types of subquestions written about the correct (selective) or incorrect (eliminative) options of their main questions, respectively. The question and option lengths of the main questions are separately reported in parentheses for comparison. *"None"* denotes "None of the above choices."

sive evaluation of the models tested on ReClor, we use all of our main questions and subquestions as a test set. Our hypothesis is that if the models can effectively generalize to understand the rationale behind the correct answer, they should exhibit a similar degree of performance on both the main questions and subquestions.

**Evaluation Metrics** In addition to the simple accuracy over the main questions (*MainQ Accuracy*) and subquestions (*SubQ Accuracy*), we calculate the accuracy across the subquestions written for the correct and incorrect original options (*Selective* and *Eliminative SubQ Accuracy*), respectively. We also calculate the *Consistency* score to see how often a model answers both the main question and all of its subquestions correctly and thereby shows the comprehensive capability of critical reasoning. Because the SubQ accuracy is a micro average, we also report a macro average for reference (*MainQ-wise SubQ Accuracy*). To compute these scores for humans, we take a majority vote of the three labels for each main question and subquestion.

### 4.1 Models and Settings

The models we evaluate are either in the fully-finetuned setting on the training set of ReClor (excluding our main questions), few-shot of ReClor, and zero-shot that uses only the task instruction.

**Fully-Finetuned Models** We use DeBERTa-v3 (large; He et al., 2023) and UnifiedQA-v2 (base, large, and 3B; Khashabi et al., 2020, 2022). Both

models are reported to exhibit strong generalization performance on QA datasets.

**Few- and Zero-Shot Models** We include recent LLMs such as FLAN-T5 (XXL; Chung et al., 2022), Flan-UL2 (20B; Tay et al., 2023), Vicuna (7B and 13B; Chiang et al., 2023), LLaMA 2 (7B to 70B; Touvron et al., 2023b), Mistral (7B; Jiang et al., 2023) and InstructGPT (`text-davinci-003`; Ouyang et al., 2022). In the few-shot setting, the input prompt has five ReClor exemplars. Because some models only accept a limited length of input, we only report one-shot results of those models. For reference, we report few-shot results using RULE examples. The zero-shot prompt only has the task instruction. We also include Chain-of-Thoughts (CoT; Wei et al., 2022) and zero-shot CoT (Kojima et al., 2022) of Instruct-GPT, providing the models with explanatory examples to potentially enhance their performance. In CoT, the prompt includes ReClor exemplars each of which is followed by the rationale of the correct answer option that is collected in this study. Appendix D shows examples of our CoT prompt.

In the few- and zero-shot settings, we follow the test split approach used by Ravichander et al. (2022) and split our dataset into five disjoint sets to measure the variability of models' performance. Appendix E describes the details.

### 4.2 Results

Table 2 presents our main results. In the fully-finetuned setting, we observe that the SubQ accuracy does not significantly exceed the chance rate (25.0%), which is far below the zero-shot performance of UnifiedQA-v2 as well as the human performance. This degradation may be due to overfitting to ReClor examples, by which the models rely heavily on superficial features of answer options that are not useful in answering the subquestions. In our dataset, a group of subquestions shares the same set of four rationales, which requires that the models closely examine the question texts.

In the few- and zero-shot settings, we observe that the highest accuracy is 80.3% on the main questions by LLaMA 2 70B with five-shot exemplars of ReClor and 65.7% on the subquestions by Flan-UL2 in the zero-shot setting. Both the MainQ and the SubQ accuracies are lower than the human accuracy by large margins ($\Delta = 11.2\%, 16.9\%$), highlighting a severe limitation in the models' rationale understanding; in most cases, the models may

only understand part of the necessary rationales for the comprehension process.

Although it is not our intended task setting, when we use a part of the subquestions for in-context learning, the highest SubQ accuracy is 70.1% by InstructGPT in the five-shot setting. This result is still below the human accuracy by a noticeable margin. Interestingly, the in-context learning on subquestions is not helpful for smaller models such as Vicuna 7B and 13B.

Looking at the best Selective and Eliminative SubQ Accuracies, we find that although the former accuracy (five-shot LLaMA 2 70B, 90.0%) is close to the human performance, the latter accuracy (zero-shot Flan-UL2, 59.1%) is significantly below the human performance (78.9%). This contrast shows that answering the eliminative subquestions is difficult for the models, highlighting the limited capacity of LLMs: Even if the models can choose the correct answer option, they may not understand why incorrect answer options should be refuted.

Consistency and MainQ-wise SubQ Accuracy also conform to this trend. Although the consistency by humans is not high (52.9%), probably owing to the difficulty of the subquestions, a large margin still exists between the human consistency and the best consistency by InstructGPT (18.2%). MainQ-wise SubQ Accuracy provides a bit more intuitive observation: The best model answers only 64.3% of the subquestions per one main question, although humans get them wrong less often (81.5%). We report the detailed number of MainQ-wise SubQ Accuracy in Appendix F.

Contrary to our expectations, CoT does not improve the performance of InstructGPT. Rather, it leads to a decline in the MainQ and SubQ accuracies. This result is consistent with findings on the unreliable nature of CoT (Wang et al., 2023; Turpin et al., 2023), which may be exposed by the complexity of critical reasoning.

**Does the Model Answer "None of the above choices" Questions Correctly?** Some of our subquestions contain "None of the above choices," which might make the questions challenging. In particular, the model performance on this type of question might be strongly affected by the in-context learning of exemplars. To investigate this hypothesis, we calculate the accuracy of the subquestions that include the "None" option. In the five-shot InstructGPT using RULE examples, we find that although the model achieves 62.7% ac-

| Model | # Param | MainQ Acc. | SubQ Acc. | Selective SubQ Acc. | Eliminative SubQ Acc. | Consist. | MainQ-wise SubQ Acc. |
|---|---|---|---|---|---|---|---|
| *Fully Finetuned on ReClor* | | | | | | | |
| DEBERTA-v3-LARGE | 304M | 66.0 | **33.1** | **56.1** | **25.0** | **2.4** | **32.8** |
| UNIFIEDQA-v2-BASE | 220M | 40.5 | 25.8 | 21.3 | 27.4 | 0.7 | 26.0 |
| UNIFIEDQA-v2-LARGE | 770M | 57.7 | 25.0 | 19.9 | 26.8 | 1.4 | 24.7 |
| UNIFIEDQA-v2-3B | 3B | **66.8** | 25.3 | 21.8 | 26.6 | 1.4 | 25.2 |
| *Five-Shot on ReClor* | | | | | | | |
| VICUNA 13B | 13B | $46.2_{\pm 0.7}$ | $50.0_{\pm 4.4}$ | $78.2_{\pm 3.0}$ | $40.1_{\pm 4.6}$ | 5.6 | 49.4 |
| FLAN-UL2 | 20B | $58.5_{\pm 0.3}$ | $\mathbf{65.5}_{\pm 5.1}$ | $88.0_{\pm 4.0}$ | $\mathbf{57.6}_{\pm 5.4}$ | 16.9 | **64.3** |
| INSTRUCTGPT | N/A | $71.8_{\pm 1.0}$ | $65.3_{\pm 1.8}$ | $88.4_{\pm 2.5}$ | $57.1_{\pm 1.5}$ | **18.2** | 64.0 |
| INSTRUCTGPT + CoT | N/A | $67.8_{\pm 0.5}$ | $63.2_{\pm 2.1}$ | $88.5_{\pm 2.5}$ | $54.2_{\pm 2.8}$ | 17.2 | 61.8 |
| LLAMA2 13B | 13B | $48.5_{\pm 2.5}$ | $44.6_{\pm 3.2}$ | $75.3_{\pm 3.4}$ | $33.8_{\pm 4.0}$ | 5.3 | 44.7 |
| LLAMA2 70B | 70B | $\mathbf{80.3}_{\pm 0.4}$ | $60.0_{\pm 2.6}$ | $\mathbf{90.0}_{\pm 1.1}$ | $49.4_{\pm 2.9}$ | 17.7 | 59.3 |
| MISTRAL 7B | 7B | $59.9_{\pm 0.9}$ | $55.3_{\pm 3.4}$ | $83.6_{\pm 3.4}$ | $45.4_{\pm 3.6}$ | 9.0 | 54.4 |
| *Five-Shot on RULE (for reference)* | | | | | | | |
| VICUNA 13B | 13B | $43.9_{\pm 1.3}$ | $44.2_{\pm 2.7}$ | $72.6_{\pm 2.6}$ | $34.2_{\pm 2.6}$ | 4.1 | 44.0 |
| FLAN-UL2 | 20B | $57.9_{\pm 0.2}$ | $66.0_{\pm 4.9}$ | $87.7_{\pm 4.6}$ | $58.4_{\pm 5.0}$ | 17.8 | 64.9 |
| INSTRUCTGPT | N/A | $70.2_{\pm 0.4}$ | $\mathbf{70.1}_{\pm 2.3}$ | $90.0_{\pm 3.5}$ | $\mathbf{63.0}_{\pm 2.0}$ | **23.1** | **69.2** |
| LLAMA2 13B | 13B | $47.7_{\pm 3.0}$ | $46.3_{\pm 4.0}$ | $80.0_{\pm 2.1}$ | $34.4_{\pm 4.7}$ | 5.1 | 47.1 |
| LLAMA2 70B | 70B | $\mathbf{78.9}_{\pm 0.6}$ | $64.0_{\pm 4.8}$ | $\mathbf{90.6}_{\pm 2.5}$ | $54.6_{\pm 5.5}$ | 21.1 | 63.5 |
| MISTRAL 7B | 7B | $58.2_{\pm 1.6}$ | $57.5_{\pm 5.4}$ | $88.1_{\pm 3.0}$ | $46.7_{\pm 7.3}$ | 9.4 | 57.2 |
| *Zero-Shot* | | | | | | | |
| UNIFIEDQA-v2-3B | 3B | 45.5 | $47.9_{\pm 2.1}$ | $71.6_{\pm 2.9}$ | $39.4_{\pm 2.2}$ | 5.7 | 47.8 |
| UNIFIEDQA-v2-11B | 11B | 55.2 | $57.3_{\pm 2.7}$ | $74.8_{\pm 5.2}$ | $51.1_{\pm 2.7}$ | 9.7 | 56.5 |
| FLAN-T5-XXL | 11B | 60.0 | $64.3_{\pm 4.0}$ | $86.2_{\pm 5.4}$ | $56.5_{\pm 3.3}$ | 14.7 | 63.4 |
| VICUNA 13B | 13B | 44.2 | $49.5_{\pm 2.7}$ | $77.1_{\pm 1.7}$ | $39.7_{\pm 2.7}$ | 6.2 | 49.4 |
| FLAN-UL2 | 20B | 56.2 | $\mathbf{65.7}_{\pm 5.2}$ | $84.5_{\pm 4.4}$ | $\mathbf{59.1}_{\pm 5.1}$ | 14.7 | **64.2** |
| INSTRUCTGPT | N/A | 64.1 | $62.8_{\pm 2.2}$ | $\mathbf{89.9}_{\pm 2.0}$ | $53.2_{\pm 2.1}$ | **15.5** | 61.8 |
| INSTRUCTGPT+ CoT | N/A | 63.8 | $62.3_{\pm 1.0}$ | $89.6_{\pm 1.5}$ | $52.6_{\pm 1.5}$ | 14.2 | 61.2 |
| LLAMA2 13B | 13B | 43.8 | $44.4_{\pm 3.0}$ | $75.3_{\pm 3.1}$ | $33.5_{\pm 2.5}$ | 4.7 | 44.5 |
| LLAMA2 70B | 70B | **70.8** | $58.0_{\pm 3.7}$ | $88.1_{\pm 3.4}$ | $47.3_{\pm 4.0}$ | 14.1 | 57.3 |
| MISTRAL 7B | 7B | 54.0 | $55.9_{\pm 3.2}$ | $85.9_{\pm 2.0}$ | $45.3_{\pm 3.6}$ | 8.6 | 55.0 |
| HUMAN | - | **91.5** | **82.6** | **93.0** | **78.9** | **52.9** | **81.5** |

Table 2: Model performance on our RULE dataset consisting of the main questions (*MainQ*) and subquestions (*SubQ*). We report the accuracy for the subquestions written about the correct option (*Selective SubQ Acc.*) and incorrect options (*Eliminative SubQ Acc.*) of the main questions. Consistency holds only when the model answers both the main question and its subquestions correctly. InstructGPT is `text-davinci-003`.

| Batch | Acc. | # *None* in shot | *None* Acc. |
|---|---|---|---|
| #1 | 70.1 | 0 | 10.3 |
| #2 | 69.7 | 0 | 25.9 |
| #3 | 72.9 | 0 | 0.0 |
| #4 | 71.3 | 1 | 43.8 |
| #5 | 66.0 | 1 | 40.6 |
| Avg. | 70.1 | 0.4 | 32.0 |

Table 3: Accuracy of the subquestions that have "None of the above choices" as the correct answer (*None Acc*), compared to that of all subquestions (*Acc*). *None in shot* indicates how many "None" examples are included in the few-shot prompt for each test split.

curacy for the subquestions that have the "None" option, it shows 32.0% when "None" is the correct answer. This low accuracy is decomposed into 40.9% accuracy if the prompt includes the "None"

option as the correct answer and 13.7% accuracy otherwise. These results demonstrate that using exemplars helps to answer those questions to some extent but not significantly. Table 3 reports the accuracy of five-shot InstructGPT across the five batches.

We report the complementary results of the main experiment in Appendix G, in which the one-shot setting does not improve the model performance consistently. Appendix H shows the SubQ accuracy only for the main questions the models answer correctly. Appendix I shows the performance plot across the question and option length.

## 5 Analysis

To qualitatively investigate the models' behavior observed in Section 4, we aim to answer the following research questions.

**Why Are the Eliminative Subquestions Difficult?** As discussed in the previous section, we find a performance discrepancy between the selective and eliminative subquestions. We attribute this discrepancy to two potential reasons. First, the eliminative subquestions are inherently complex because of the negation included in their question text, which the models may find difficult to handle (Ravichander et al., 2022). Second, the model may lack the ability to comprehend why certain options are incorrect, which is partially supported by studies that highlight the susceptibility for distractors in the multiple-choice QA (Si et al., 2021).

To distinguish between the difficulty of comprehending complex questions and that of refuting relevant alternatives in the eliminative subquestions, we develop a follow-up task, *rationale alignment*. In this task, given a context, the main question, one of the main options, and four rationales, the model selects one out of the four rationales that validates the correctness of the given option. We use Instruct-GPT in the five-shot setting and report the average results from five different prompts. Appendix J provides the input prompt.

Because the subquestion text is not used in this task, we expect that the results are not affected by the complexity of subquestion texts. The result is 89.7% and 31.5% accuracy for the correct and incorrect answer options, respectively, showing a distinct difference between them. This discrepancy suggests the model's serious deficiency in comprehending eliminative rationales.

**Is the Model Better at Writing Rationales than Humans?** Given that CoT does not improve the model performance, we are interested in the quality and potential usefulness of model-generated rationales compared to our human-written rationales. We use a similar prompt to that used in our CoT setting, instructing InstructGPT to generate rationales for 50 options. We then randomly shuffle the order of human-written and model-generated rationales, and manually annotate which rationale is better in terms of necessity and specificity. The result is 35 wins by humans and 15 wins by the model among the 50 comparisons, showing that the human-written rationales are likely to be more detailed and supportive than the model-generated ones. In particular, we find that the model rationales struggle to capture the *implicit* rationale necessary for certifying the validity of the target option. When the rationale is explicit and described well

|  | Direct | Indirect | Total |
|---|---|---|---|
| Contextual | 37 / 47 | 28 / 22 | 65 / 69 |
| External | 22 / 20 | 13 / 11 | 35 / 31 |
| Total | 59 / 67 | 41 / 33 | 100 |

Table 4: Annotation results of rationale types on 100 examples randomly sampled from all subquestions (left) and from the error examples by InstructGPT (right).

in the context, the model rationale looks convincing and close to the human rationale. Among the 15 examples where humans lose, we find five examples unsatisfactory to validate the target option, implying that approximately 10% of unreasonable rationales are potentially included in our dataset.

**What Types of Reasoning are Required in the Rationale Understanding?** To qualitatively analyze the collected rationales, we first sample 100 subquestions to annotate reasoning types. We define two dichotomies: *direct/indirect* and *contextual/external*. Direct reasoning occurs if a rationale involves an explicit description for the certification of a target option's (in)validity, whereas indirect reasoning only provides relevant facts for the validity. Context reasoning includes facts (or their interpretation and summarization) described in the context, while external reasoning is pertinent to commonsense and norms that are not described in the context. For comparative error analysis, we also sample 100 subquestions among those that InstructGPT answers incorrectly.

We report our annotation results in Table 4. The number of the direct and contextual rationales is the largest among the other types, which further increases when we look at the error cases of InstructGPT. We find that our dataset covers a sufficient number of indirect and external reasoning, i.e., various modes of rationale understanding. Error examples for the four reasoning types are reported in Appendix K. Although we also examine the reasoning types originally labeled in the ReClor dataset, we do not observe any remarkable trends in the subquestion accuracy (Appendix L).

**Do the Rationales Help the Model to Answer the Main Questions?** Because the collected rationales are expected to support the decision of selecting and eliminating answer options, we investigate whether adding the rationales to the main questions improves the performance in the five-shot Instruct-

| Input | Accuracy |
|---|---|
| Context | 72.2 |
| + Selective Rationale | 91.4 |
| + Eliminative Rationale | 66.0 |
| + Both | 89.6 |

Table 5: MainQ accuracy of InstructGPT that uses the selective or eliminative rationales in the input.

GPT. We append the rationale to the context, main question, and four options with the `Rationale:` label. The results are shown in Table 5. We observe an improvement when the selective rationale is added; however, degradation occurs when we add the eliminative rationale, even if it is provided with the selective rationale. This result adds insight to the observation by Sun et al. (2022), showing that the model cannot use eliminative rationales for answering main questions and becomes confused by those rationales. We also investigate the context-ablated setting in Appendix M.

## 6 Conclusion

We construct a dataset to evaluate the models' ability of critical reasoning in logical reading comprehension. We crowdsource free-form rationale for main questions taken from an existing dataset and use an LLM to generate subquestion texts. Resulting questions ask about the underlying rationales for why a certain answer option should be selected and the others should be eliminated. We find that LLMs are particularly bad at answering eliminative subquestions, highlighting that those models do not necessarily have the comprehensive ability of critical reasoning. For future work, we will develop a more efficient pipeline for data collection and facilitate better rationale generation by LLMs.

## Ethical Consideration

We use crowdsourcing in our data collection. We make sure to be responsible to the crowdworkers and to make fair compensation for their work. We do not collect any personal information other than worker IDs on the platform, which are removed in our data release. Before the workers accept our tasks, we inform them of our purpose for the data collection. This study is approved by the internal review board of the authors' institutes.

## Limitations

We recognize the following limitations in this study.

**Task Format**    In this study, we focus on the multiple-choice QA task. This task format allows us to flexibly ask about various linguistic phenomena and human reasoning by selecting and eliminating alternatives, and we consider solving such a discriminative task would be a minimal requirement for human-like linguistic behaviors. However, it has an inherent limitation in assessing the ability of natural language understanding. For example, we cannot evaluate the models' ability to produce an intended output.

**Annotation Analysis**    We conduct the annotation analysis in Section 5, in which we define the reasoning types and manually review the sampled examples. Although we make our annotation data and guideline publicly available for ensuring the reproducibility of annotation results, the results of our annotation analysis inevitably involve our subjective judgments.

**Source Dataset**    We create our auxiliary questions on top of an existing English logical reading comprehension dataset, ReClor. Although our methodology of the data collection (i.e., writing the rationale for selecting and eliminating alternatives) is widely applicable to other datasets and languages, using the single dataset in the single language would limit the generalizability of our findings.

## Acknowledgments

We would like to thank the anonymous reviewers for their helpful comments. This work was supported by JST PRESTO Grant Number JP-MJPR20C4 and JSPS KAKENHI Grant Number 22K17954.

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

# A Crowdsourcing Instructions and Examples

Figures 16 to 23 show the instructions and examples we present to the crowdworkers. Figures 16 to 20 illustrate the rationale writing task, Figures 21 and 22 illustrate the rationale validation task, and Figure 23 illustrates the human validation task.

## B  Question Generation Prompt

Figure 14 shows an example of our prompt used for generating subquestions in Section 3.3.

## C  Crowdsourcing Details

To access a pool of crowdworkers, we used Amazon Mechanical Turk. The crowdworkers who took the qualification test are based in the United States, United Kingdom, or Canada, have an approval rate of at least 98%, and have at least 1,000 approved tasks. We ensure that the average payments exceed $12.00 USD per hour for each task. The rationale writing task costs $2.00 per main question (estimating that it takes seven to ten minutes to write the rationales), the rationale validation task costs $0.30 per rationale (one minute), and the human validation task $1.50 per five questions (five minutes). The rationale writing tasks, rationale validation tasks, QA validation tasks, and human performance tasks are taken by 48, 39, 52, and 24 workers, respectively. We use the crowdsourcing tool used in Nangia et al. (2021).

## D  Chain-of-Thought Prompt

Figure 15 shows an example of the prompt used in our chain-of-thought experiment. We insert the rationale between the `Answer:` label and the correct option label, with an expectation that it would help the model (InstructGPT) select the correct option.

## E  Test Split Setting

The in-context learning performance of LLMs may vary depending on the exemplars of the prompt, but it incurs a high computational cost (or financial cost for proprietary models) if we repeatedly evaluate the models on the entire dataset using various sets of different exemplars to take the average performance. Because of this cost limitation, we follow the test split approach used by Ravichander et al. (2022), splitting our dataset into five disjoint sets and testing the models on each set with different exemplars to measure the performance variance across the disjoint sets. Note that we do not split the set of the main questions, because it has only 943 examples; hence, in the few-shot setting, we take the average across five runs on all main questions. In the few-shot setting using ReClor, we sample questions disjointly from its training set, whereas

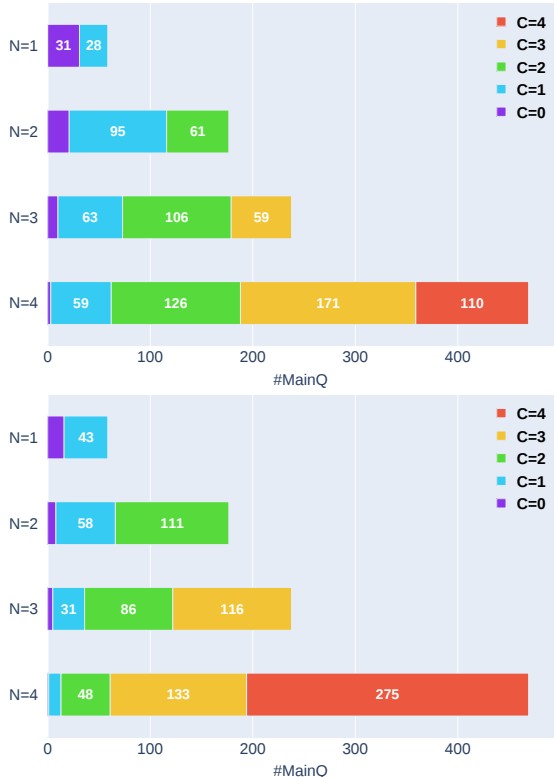

Figure 3: Distribution of correctly answered subquestions ($C$) out of the total number of subquestions ($N$), for both InstructGPT (top) and humans (bottom).

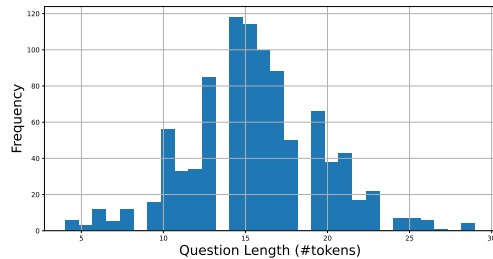

Figure 4: Distribution of the question length (#tokens) of the main questions.

in using RULE, the exemplars are sampled from the corresponding disjoint set.

## F  MainQ-wise SubQ Results of InstructGPT

Because a single main question has multiple subquestions in our dataset, we report the detailed numbers of correctly-answered SubQ by InstructGPT in Figure 3.

| Model | # Param | MainQ Acc. | SubQ Acc. | Selective SubQ Acc. | Eliminative SubQ Acc. | Consist. | MainQ-wise Acc. |
|---|---|---|---|---|---|---|---|
| *Five-Shot on ReClor* | | | | | | | |
| LLaMA 7B | 7B | $25.8_{\pm1.6}$ | $28.6_{\pm7.1}$ | $39.8_{\pm13.1}$ | $24.6_{\pm6.3}$ | 0.8 | 28.2 |
| LLaMA 13B | 13B | $38.7_{\pm2.7}$ | $36.3_{\pm3.5}$ | $63.6_{\pm4.0}$ | $26.6_{\pm3.6}$ | 2.9 | 36.6 |
| Vicuna 7B | 7B | $33.4_{\pm2.6}$ | $38.6_{\pm3.3}$ | $61.4_{\pm5.8}$ | $30.5_{\pm3.4}$ | 2.8 | 38.3 |
| LLaMA2 7B | 7B | $36.4_{\pm2.1}$ | $35.2_{\pm3.9}$ | $63.6_{\pm6.5}$ | $25.1_{\pm3.2}$ | 1.7 | 35.6 |
| *One-Shot on ReClor* | | | | | | | |
| UnifiedQA-v2-Base | 220M | $27.4_{\pm1.4}$ | $34.7_{\pm6.4}$ | $42.8_{\pm7.7}$ | $31.9_{\pm6.2}$ | 0.7 | 34.6 |
| UnifiedQA-v2-Large | 770M | $27.1_{\pm1.7}$ | $28.0_{\pm4.7}$ | $29.2_{\pm5.3}$ | $27.6_{\pm4.9}$ | 0.0 | 27.6 |
| UnifiedQA-v2-3B | 3B | $31.3_{\pm3.2}$ | $26.2_{\pm2.8}$ | $26.0_{\pm9.8}$ | $26.2_{\pm1.4}$ | 0.3 | 26.1 |
| UnifiedQA-v2-11B | 11B | $44.1_{\pm5.9}$ | $37.2_{\pm7.4}$ | $53.8_{\pm16.6}$ | $31.4_{\pm4.4}$ | 2.0 | 36.5 |
| Flan-T5-XXL | 11B | $61.3_{\pm0.3}$ | $63.7_{\pm3.7}$ | $85.9_{\pm2.6}$ | $55.9_{\pm3.9}$ | 14.0 | 62.5 |
| Flan-UL2 | 20B | $58.0_{\pm0.5}$ | $66.0_{\pm5.3}$ | $87.4_{\pm3.9}$ | $58.5_{\pm5.7}$ | 17.7 | 65.1 |
| LLaMA 7B | 7B | $26.6_{\pm0.9}$ | $32.4_{\pm4.2}$ | $50.7_{\pm11.1}$ | $26.1_{\pm3.7}$ | 1.6 | 32.5 |
| LLaMA 13B | 13B | $32.7_{\pm2.4}$ | $33.8_{\pm1.9}$ | $56.0_{\pm5.8}$ | $26.0_{\pm2.5}$ | 1.4 | 34.1 |
| LLaMA 33B | 33B | $56.1_{\pm1.0}$ | $49.3_{\pm4.3}$ | $80.1_{\pm3.8}$ | $38.5_{\pm4.3}$ | 7.4 | 49.6 |
| LLaMA 65B | 65B | $65.2_{\pm1.4}$ | $52.6_{\pm3.4}$ | $85.2_{\pm1.3}$ | $41.1_{\pm4.3}$ | 9.4 | 51.9 |
| Vicuna 7B | 7B | $35.8_{\pm2.0}$ | $38.9_{\pm2.2}$ | $65.3_{\pm3.1}$ | $29.5_{\pm2.3}$ | 1.8 | 38.1 |
| Vicuna 13B | 13B | $42.5_{\pm0.8}$ | $45.2_{\pm3.1}$ | $72.1_{\pm3.3}$ | $35.8_{\pm4.0}$ | 4.2 | 45.1 |
| InstructGPT | N/A | $67.8_{\pm0.5}$ | $64.6_{\pm1.9}$ | $87.8_{\pm2.0}$ | $56.3_{\pm1.8}$ | 17.5 | 63.6 |
| InstructGPT + CoT | N/A | $64.3_{\pm2.5}$ | $64.3_{\pm2.4}$ | $88.8_{\pm1.3}$ | $55.7_{\pm2.5}$ | 15.4 | 62.5 |
| LLaMA2 7B | 7B | $35.0_{\pm1.1}$ | $34.8_{\pm2.4}$ | $61.7_{\pm4.2}$ | $25.2_{\pm2.6}$ | 2.2 | 35.2 |
| LLaMA2 13B | 13B | $46.4_{\pm2.4}$ | $43.7_{\pm2.9}$ | $72.6_{\pm3.6}$ | $33.4_{\pm2.8}$ | 4.7 | 43.6 |
| LLaMA2 70B | 70B | $77.2_{\pm0.2}$ | $61.3_{\pm0.7}$ | $90.0_{\pm1.4}$ | $51.2_{\pm1.2}$ | 20.0 | 61.0 |
| Mistral 7B | 7B | $52.6_{\pm1.4}$ | $53.4_{\pm2.3}$ | $81.9_{\pm2.8}$ | $43.4_{\pm2.7}$ | 7.4 | 52.7 |
| *Five-Shot on RULE (for reference)* | | | | | | | |
| LLaMA 7B | 7B | $29.1_{\pm2.3}$ | $34.9_{\pm2.5}$ | $64.3_{\pm4.3}$ | $24.5_{\pm2.4}$ | 1.5 | 35.5 |
| LLaMA 13B | 13B | $36.8_{\pm3.5}$ | $35.6_{\pm2.5}$ | $68.1_{\pm3.0}$ | $24.2_{\pm2.8}$ | 2.4 | 36.0 |
| Vicuna 7B | 7B | $35.0_{\pm1.1}$ | $39.9_{\pm3.9}$ | $60.2_{\pm8.9}$ | $32.7_{\pm4.4}$ | 3.2 | 39.8 |
| LLaMA2 7B | 7B | $37.8_{\pm1.1}$ | $32.0_{\pm4.4}$ | $62.3_{\pm6.5}$ | $21.1_{\pm3.4}$ | 1.5 | 32.4 |
| *One-Shot on RULE (for reference)* | | | | | | | |
| UnifiedQA-v2-Base | 220M | $27.7_{\pm2.7}$ | $36.5_{\pm2.5}$ | $38.5_{\pm4.0}$ | $35.8_{\pm2.5}$ | 1.6 | 36.9 |
| UnifiedQA-v2-Large | 770M | $28.3_{\pm1.4}$ | $27.4_{\pm1.2}$ | $27.4_{\pm10.2}$ | $27.4_{\pm3.0}$ | 1.0 | 27.7 |
| UnifiedQA-v2-3B | 3B | $35.0_{\pm1.4}$ | $30.1_{\pm4.8}$ | $35.8_{\pm10.7}$ | $28.2_{\pm4.0}$ | 2.0 | 30.6 |
| UnifiedQA-v2-11B | 11B | $42.6_{\pm7.1}$ | $37.4_{\pm11.7}$ | $47.5_{\pm17.7}$ | $33.9_{\pm9.7}$ | 3.1 | 38.0 |
| Flan-T5-XXL | 11B | $60.6_{\pm0.5}$ | $64.1_{\pm3.6}$ | $85.8_{\pm3.1}$ | $56.5_{\pm3.9}$ | 14.0 | 63.2 |
| Flan-UL2 | 20B | $57.6_{\pm0.4}$ | $66.0_{\pm4.5}$ | $87.4_{\pm3.8}$ | $58.5_{\pm4.7}$ | 17.2 | 64.9 |
| LLaMA 7B | 7B | $28.2_{\pm2.4}$ | $32.9_{\pm2.5}$ | $48.3_{\pm8.1}$ | $27.4_{\pm4.1}$ | 1.6 | 33.6 |
| LLaMA 13B | 13B | $30.0_{\pm3.2}$ | $32.9_{\pm1.3}$ | $50.9_{\pm4.3}$ | $26.6_{\pm1.8}$ | 1.7 | 33.4 |
| LLaMA 33B | 33B | $53.3_{\pm3.2}$ | $48.4_{\pm4.5}$ | $79.7_{\pm3.4}$ | $37.3_{\pm4.7}$ | 6.0 | 48.4 |
| LLaMA 65B | 65B | $62.8_{\pm3.4}$ | $52.8_{\pm6.0}$ | $84.9_{\pm4.4}$ | $41.5_{\pm6.5}$ | 7.5 | 52.0 |
| Vicuna 7B | 7B | $34.7_{\pm1.8}$ | $41.2_{\pm2.4}$ | $65.3_{\pm6.4}$ | $32.6_{\pm3.3}$ | 3.4 | 41.2 |
| Vicuna 13B | 13B | $40.7_{\pm2.3}$ | $41.7_{\pm1.6}$ | $67.3_{\pm4.8}$ | $32.7_{\pm1.8}$ | 3.5 | 41.8 |
| InstructGPT | N/A | $65.4_{\pm1.9}$ | $66.5_{\pm1.3}$ | $89.0_{\pm0.8}$ | $58.5_{\pm1.2}$ | 19.5 | 65.5 |
| InstructGPT + CoT | N/A | $64.3_{\pm2.5}$ | $64.3_{\pm2.4}$ | $88.8_{\pm1.3}$ | $55.7_{\pm2.5}$ | 15.4 | 62.5 |
| LLaMA2 7B | 7B | $32.8_{\pm3.3}$ | $33.4_{\pm1.7}$ | $56.1_{\pm1.8}$ | $25.4_{\pm1.9}$ | 1.4 | 34.0 |
| LLaMA2 13B | 13B | $45.0_{\pm5.5}$ | $40.9_{\pm2.9}$ | $70.6_{\pm4.4}$ | $30.3_{\pm2.6}$ | 2.8 | 41.1 |
| LLaMA2 70B | 70B | $76.0_{\pm0.2}$ | $61.5_{\pm2.8}$ | $90.4_{\pm3.1}$ | $51.4_{\pm2.7}$ | 19.2 | 61.1 |
| Mistral 7B | 7B | $50.8_{\pm1.1}$ | $53.8_{\pm3.3}$ | $84.0_{\pm3.7}$ | $43.2_{\pm3.6}$ | 6.5 | 54.0 |
| *Zero-Shot* | | | | | | | |
| UnifiedQA-v2-Base | 220M | 30.4 | $42.2_{\pm1.0}$ | $48.5_{\pm2.9}$ | $39.9_{\pm1.3}$ | 2.7 | 41.7 |
| UnifiedQA-v2-Large | 770M | 41.4 | $42.9_{\pm1.8}$ | $55.0_{\pm4.9}$ | $38.5_{\pm1.3}$ | 3.3 | 41.9 |
| LLaMA 7B | 7B | 27.7 | $27.4_{\pm4.1}$ | $38.2_{\pm4.2}$ | $23.6_{\pm4.6}$ | 0.8 | 27.1 |
| LLaMA 13B | 13B | 31.7 | $36.0_{\pm3.3}$ | $59.3_{\pm2.1}$ | $27.8_{\pm3.7}$ | 1.4 | 36.7 |
| Vicuna 7B | 7B | 36.7 | $40.5_{\pm2.3}$ | $73.1_{\pm3.2}$ | $28.9_{\pm2.9}$ | 2.4 | 40.1 |
| LLaMA2 7B | 7B | 32.3 | $35.9_{\pm3.9}$ | $66.4_{\pm4.7}$ | $25.1_{\pm3.4}$ | 2.5 | 36.3 |
| Human | - | 91.5 | 82.6 | 93.0 | 78.9 | 52.9 | 81.5 |

Table 6: Complementary results of the model performance on our dataset including the models in the one-shot setting and omitting those in the five-shot and zero-shot settings.

## G Complementary Few-Shot and Zero-Shot Results

In Table 6, we report the complementary results of few-shot settings, including the models on the one-

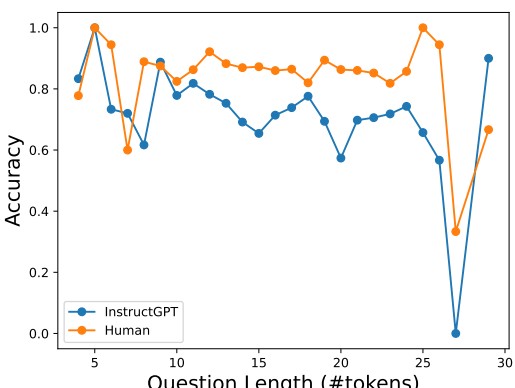

Figure 5: Comparison between the model and human accuracy and question length for the main questions.

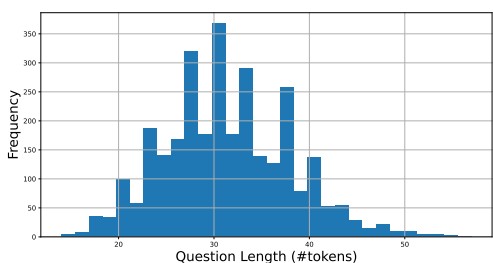

Figure 6: Distribution of the question length (#tokens) of the subquestions.

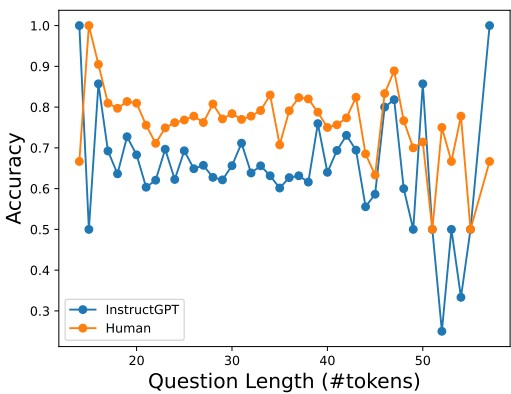

Figure 7: Comparison between the model and human accuracy and question length for the subquestions.

| Instruction | Corr. Opt. | Incorr. Opt. |
|---|---|---|
| Yes | 89.7 (784) | 31.5 (2,218) |
| No | 88.7 (781) | 28.1 (2,213) |

Table 7: Result of the rationale alignment task with and without the task instruction.

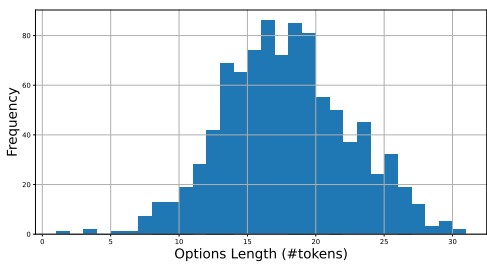

Figure 8: Distribution of the option length (#tokens) of the main questions.

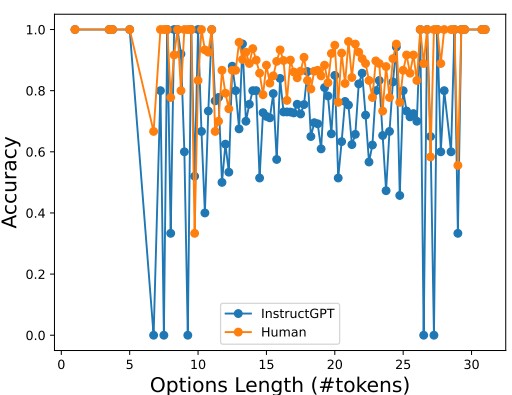

Figure 9: Comparison between the model and human accuracy and option length for the main questions.

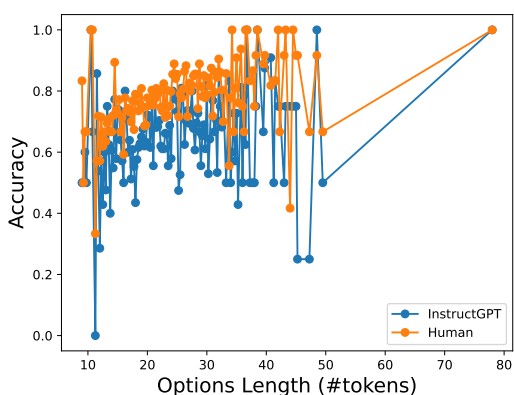

Figure 10: Comparison between the model and human accuracy and option length for the subquestions.

shot setting. We also report the results of LLaMA (7B to 65B; Touvron et al., 2023a) for reference.

## H Main Results of the Subquestions for the Correctly-Answered Main Questions

Table 8 shows the main results of the model performance on the subquestions for the main questions

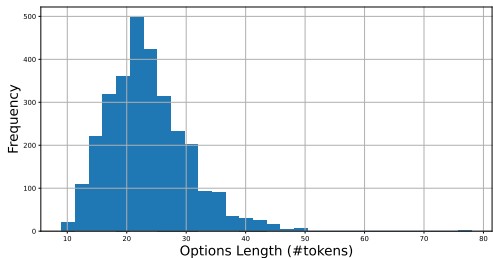

Figure 11: Distribution of the option length (#tokens) of the subquestions.

that are correctly answered by the model. Overall, we observe similar trends to the main results with the standard SubQ accuracy. Interestingly, the models' SubQ accuracies do not significantly improve even when we focus only on the correctly-answered main questions.

## I   Relationship between Question and Option Length and Model Performance

In Figures 4 to 10, we plot the distribution of the questions and options length and the model performance according to those lengths.

## J   Rationale Alignment Task

In the rationale alignment task, we test Instruct-GPT in the five-shot setting. Similar to the main experiment, we report the average results from five prompts. Each prompt is composed of five exemplars, with two exemplars presenting the correct option and three exemplars presenting the incorrect option. Figure 12 shows an example of our prompt.

The results with and without the task instruction are shown in Table 7. The performance gap between the correct and incorrect options implies that such advanced models may simply infer the correct answer without properly discriminating against incorrect options. Such a situation raises two issues: (1) the inability to reason logically like a human, and (2) the limitations of ability measurement using distractors. The first issue suggests that the model may not be able to make a clear distinction between what is correct and what is incorrect. The second issue is that the alternatives in the multiple-choice QA task are generally expected to distinguish between test takers with and without sufficient knowledge (Gierl et al., 2017), but such an expectation may not be met in our dataset.

## K   Reasoning Type Annotation

Table 10 shows examples of reasoning types we define in the annotation analysis. See Table 11 for a full example that has the main question and two subquestions.

## L   ReClor Reasoning Types and Subquestion Accuracy

Figure 13 shows the relation between the subquestion accuracy and the reasoning types defined in the original ReClor dataset. Although we do not observe significant performance differences, we see higher accuracy in Match Structures, Evaluation, Strengthen, and Weaken reasoning, and lower accuracy in Sufficient Assumptions, Technique, and Role reasoning.

## M   Context-Ablation Analysis

We try to answer the question "Does the context help in answering subquestions?" in the context-ablation setting. By removing the context, we analyze the model performance on the subquestions (and the main questions for reference) to see the dependency between question texts and answer options. The results in Table 9 show the performance reduction by approximately 4 points in the zero-shot setting and no reduction in the five-shot setting. This result implies question texts depend on answer options to some extent, which potentially makes the subquestions difficult for the models, given the first analysis in this section.

## N   Similarity of Rationale with MainQ Option

In our process to validate specificity, even if a rationale has the same meaning as the MainQ's option, we can not exclude it. This implies that some rationales might have the similar meaning as the option and not serve as a valid rationale. To examine this potential issue, we sample 50 random questions from both the selective SubQ and the eliminative SubQ. We then count how many of these rationales are semantically similar to the MainQ's option. We found three such instances in the selective SubQ and one in the eliminative SubQ, which are shown in Table 12.

| Model | # Param | MainQ Acc. | SubQ Acc. | Selective SubQ Acc. | Eliminative SubQ Acc. | Consist. | MainQ-wise SubQ Acc. |
|---|---|---|---|---|---|---|---|
| *Fully Finetuned on ReClor* | | | | | | | |
| DeBERTa-v3-Large | 304M | 66.0 | 33.1 | 60.4 | 22.5 | 3.7 | 32.2 |
| UnifiedQA-v2-Base | 220M | 40.5 | 25.8 | 19.7 | 26.7 | 1.8 | 25.0 |
| UnifiedQA-v2-Large | 770M | 57.7 | 25.0 | 17.3 | 27.7 | 2.4 | 24.6 |
| UnifiedQA-v2-3B | 3B | 66.8 | 25.3 | 19.9 | 25.7 | 2.1 | 24.1 |
| *Five-Shot on ReClor* | | | | | | | |
| Flan-UL2 | 20B | $58.5_{\pm0.3}$ | $66.3_{\pm6.3}$ | $89.6_{\pm4.6}$ | $57.2_{\pm5.1}$ | 28.8 | 65.1 |
| LLaMA 7B | 7B | $25.8_{\pm1.6}$ | $28.5_{\pm7.5}$ | $34.3_{\pm12.2}$ | $24.2_{\pm5.7}$ | 3.4 | 27.2 |
| LLaMA 13B | 13B | $38.7_{\pm2.7}$ | $37.0_{\pm4.0}$ | $65.0_{\pm6.7}$ | $25.9_{\pm3.8}$ | 7.4 | 37.2 |
| LLaMA 33B | 33B | $58.5_{\pm1.2}$ | $47.7_{\pm4.2}$ | $77.5_{\pm6.3}$ | $38.3_{\pm5.8}$ | 10.6 | 47.3 |
| LLaMA 65B | 65B | $69.1_{\pm0.9}$ | $54.4_{\pm1.7}$ | $85.5_{\pm2.8}$ | $46.9_{\pm5.6}$ | 16.0 | 54.1 |
| Vicuna 7B | 7B | $33.4_{\pm2.6}$ | $40.0_{\pm5.1}$ | $62.7_{\pm6.9}$ | $30.0_{\pm3.0}$ | 8.5 | 39.7 |
| Vicuna 13B | 13B | $46.2_{\pm0.7}$ | $48.5_{\pm6.1}$ | $76.0_{\pm8.8}$ | $42.2_{\pm5.4}$ | 12.1 | 48.1 |
| InstructGPT | N/A | $71.8_{\pm1.0}$ | $64.1_{\pm3.5}$ | $89.2_{\pm3.8}$ | $60.7_{\pm6.1}$ | 25.1 | 62.8 |
| InstructGPT + CoT | N/A | $67.8_{\pm0.5}$ | $62.9_{\pm3.1}$ | $89.5_{\pm4.2}$ | $55.8_{\pm3.6}$ | 24.9 | 61.5 |
| LLaMA2 13B | 13B | $48.5_{\pm2.5}$ | $45.8_{\pm4.2}$ | $79.1_{\pm4.0}$ | $33.1_{\pm5.2}$ | 11.1 | 45.8 |
| LLaMA2 70B | 70B | $80.3_{\pm0.4}$ | $59.9_{\pm2.5}$ | $90.7_{\pm1.9}$ | $50.5_{\pm7.2}$ | 21.8 | 59.4 |
| Mistral 7B | 7B | $59.9_{\pm0.9}$ | $54.7_{\pm4.4}$ | $84.5_{\pm2.8}$ | $46.8_{\pm2.8}$ | 15.0 | 53.7 |
| *Five-Shot on RULE (for reference)* | | | | | | | |
| Flan-UL2 | 20B | $57.9_{\pm0.2}$ | $67.2_{\pm6.2}$ | $89.4_{\pm4.4}$ | $57.1_{\pm3.6}$ | 30.9 | 66.2 |
| LLaMA 7B | 7B | $29.1_{\pm2.3}$ | $34.6_{\pm3.2}$ | $66.2_{\pm8.0}$ | $23.7_{\pm1.3}$ | 5.4 | 34.9 |
| LLaMA 13B | 13B | $36.8_{\pm3.5}$ | $35.1_{\pm3.2}$ | $67.4_{\pm5.3}$ | $24.4_{\pm2.6}$ | 6.6 | 35.0 |
| LLaMA 33B | 33B | $53.6_{\pm0.4}$ | $48.5_{\pm6.0}$ | $78.5_{\pm5.9}$ | $36.8_{\pm5.4}$ | 10.6 | 47.6 |
| LLaMA 65B | 65B | $66.2_{\pm0.7}$ | $57.2_{\pm5.1}$ | $86.4_{\pm3.1}$ | $53.7_{\pm10.3}$ | 18.3 | 56.0 |
| Vicuna 7B | 7B | $35.0_{\pm1.1}$ | $40.3_{\pm3.1}$ | $61.4_{\pm8.8}$ | $32.4_{\pm4.8}$ | 9.5 | 40.8 |
| Vicuna 13B | 13B | $43.9_{\pm1.3}$ | $43.9_{\pm2.6}$ | $73.0_{\pm3.1}$ | $34.2_{\pm3.7}$ | 9.4 | 43.4 |
| InstructGPT | N/A | $70.2_{\pm0.4}$ | $70.2_{\pm3.5}$ | $91.0_{\pm4.6}$ | $64.3_{\pm2.4}$ | 32.7 | 69.3 |
| InstructGPT + CoT | N/A | $67.8_{\pm0.5}$ | $62.9_{\pm3.1}$ | $89.5_{\pm4.2}$ | $55.8_{\pm3.6}$ | 24.9 | 61.5 |
| LLaMA2 13B | 13B | $47.7_{\pm3.0}$ | $46.9_{\pm4.7}$ | $79.4_{\pm6.3}$ | $33.6_{\pm4.6}$ | 11.0 | 47.0 |
| LLaMA2 70B | 70B | $78.9_{\pm0.6}$ | $63.7_{\pm4.2}$ | $90.8_{\pm2.5}$ | $55.2_{\pm9.2}$ | 26.5 | 63.0 |
| Mistral 7B | 7B | $58.2_{\pm1.6}$ | $56.2_{\pm5.1}$ | $88.0_{\pm3.6}$ | $49.4_{\pm7.9}$ | 16.3 | 55.6 |
| *Zero-Shot* | | | | | | | |
| UnifiedQA-v2-Base | 220M | 30.4 | $43.0_{\pm2.9}$ | $51.0_{\pm7.5}$ | $39.8_{\pm1.7}$ | 8.7 | 43.4 |
| UnifiedQA-v2-Large | 770M | 41.4 | $43.9_{\pm4.1}$ | $59.9_{\pm7.6}$ | $39.0_{\pm2.7}$ | 7.9 | 42.5 |
| UnifiedQA-v2-3B | 3B | 45.5 | $49.3_{\pm2.4}$ | $75.4_{\pm4.1}$ | $38.9_{\pm3.1}$ | 12.6 | 49.5 |
| UnifiedQA-v2-11B | 11B | 55.2 | $56.8_{\pm2.9}$ | $77.6_{\pm7.1}$ | $53.0_{\pm3.8}$ | 17.5 | 55.9 |
| Flan-T5-XXL | 11B | 60.0 | $63.2_{\pm3.4}$ | $87.3_{\pm4.8}$ | $59.2_{\pm4.6}$ | 24.6 | 62.7 |
| Flan-UL2 | 20B | 56.2 | $65.3_{\pm5.9}$ | $86.0_{\pm4.8}$ | $60.5_{\pm4.7}$ | 26.2 | 63.7 |
| LLaMA 7B | 7B | 27.7 | $27.0_{\pm4.5}$ | $37.5_{\pm2.8}$ | $23.6_{\pm3.9}$ | 3.1 | 26.9 |
| LLaMA 13B | 13B | 31.7 | $35.5_{\pm3.0}$ | $55.9_{\pm4.0}$ | $27.5_{\pm3.8}$ | 4.3 | 35.9 |
| LLaMA 33B | 33B | 54.5 | $50.2_{\pm4.6}$ | $81.8_{\pm4.3}$ | $41.4_{\pm5.5}$ | 12.5 | 50.2 |
| LLaMA 65B | 65B | 52.1 | $47.5_{\pm2.2}$ | $80.1_{\pm2.5}$ | $38.0_{\pm5.4}$ | 10.4 | 46.6 |
| Vicuna 7B | 7B | 36.7 | $40.8_{\pm2.8}$ | $77.5_{\pm2.8}$ | $29.4_{\pm2.8}$ | 6.6 | 40.3 |
| Vicuna 13B | 13B | 44.2 | $48.9_{\pm4.8}$ | $76.9_{\pm3.4}$ | $40.4_{\pm1.8}$ | 13.9 | 48.2 |
| InstructGPT | N/A | 64.1 | $61.9_{\pm2.9}$ | $89.8_{\pm2.9}$ | $55.8_{\pm3.6}$ | 24.2 | 60.8 |
| InstructGPT + CoT | N/A | 63.8 | $60.9_{\pm1.4}$ | $89.1_{\pm2.4}$ | $55.9_{\pm3.2}$ | 22.3 | 59.7 |
| LLaMA2 13B | 13B | 43.8 | $45.5_{\pm3.7}$ | $77.6_{\pm5.3}$ | $33.3_{\pm3.0}$ | 10.7 | 45.4 |
| LLaMA2 70B | 70B | 70.8 | $57.2_{\pm4.0}$ | $88.2_{\pm3.2}$ | $49.9_{\pm5.4}$ | 19.9 | 56.3 |
| Mistral 7B | 7B | 54.0 | $54.7_{\pm3.4}$ | $85.1_{\pm5.5}$ | $47.1_{\pm4.3}$ | 15.9 | 53.3 |
| Human | - | 91.5 | 82.9 | 92.8 | 79.3 | 57.8 | 81.6 |

Table 8: Main results of the model performance on our dataset focusing on the subquestions for the main questions that are correctly answered by the model.

| Setting | MainQ | SubQ | Selective SubQ | Eliminat. SubQ |
|---|---|---|---|---|
| 0-shot | $41.0_{-23.0}$ | $58.8_{-4.2}$ | $86.4_{-3.7}$ | $53.4_{-4.3}$ |
| 5-shot | $42.5_{-29.7}$ | $70.5_{+0.0}$ | $86.9_{-3.1}$ | $64.7_{+1.7}$ |

Table 9: Context-ablated accuracy. The subscript values indicate the accuracy gap against the full-input setting.

In a given context, you'll be given a question, an answer, and four rationales. Your task is to identify the rationale that explains the correctness of the provided option the best. If the option is wrong, choose the rationale that explains why it is wrong. Conversely, if the option is correct, choose the rationale that explains why it is correct.

**Context:** Teachers should not do anything to cause their students to lose respect for them. And students can sense when someone is trying to hide his or her ignorance. Therefore, a teacher who does not know the answer to a question a student has asked should not pretend to know the answer. Question: The conclusion is properly drawn if which one of the following is assumed?

**Question:** The conclusion is properly drawn if which one of the following is assumed?

**Option:** Students' respect for a teacher is independent of the amount of knowledge they attribute to that teacher.

→ **Incorrect Option**

**Rationale0:** The ranking of students' respect for honesty is not relevant to the conclusion of a teacher shouldn't pretend to know an answer to question they don't know the answer to.

**Rationale1:** The assumption is that students' respect for the teacher is based on how much knowledge the teacher has.

**Rationale2:** The conclusion is that teachers shouldn't pretend to know the answer to a question that they don't know, so the assumption is that student's respect for a teacher is interlinked to the student's perceived knowledge of the teacher.

**Rationale3:** The conclusion does not have anything to do with a teacher being effective.

**Answer:** The answer is Rationale2

**Context:** Miguel has four family members who plan to come to his graduation on Sunday afternoon, but it is likely that only three of them will be allowed to attend. Normally graduation is held in the football stadium, where there is no limit on the number of family members who can attend. However, the ceremony is relocated to the gymnasium if it rains, and each graduate receives just three admission tickets for use by family members.

**Question:** The conclusion of the argument is most strongly supported if which one of the following is assumed?

**Option:** The weather service has indicated that there is a very high likelihood of rain on Sunday afternoon.

→ **Correct Option**

**Rationale0:** No mention is made of whether un-needed spaces can be transferred between students, and so this cannot be assumed to impact the number of spaces available to Miguel's family.

**Rationale1:** Abnormally large class size may not preclude Miguel from having more than three family members attend, as the football stadium is a possible venue and has no limitation on the number who may attend.

**Rationale2:** A family member who cannot attend the graduation has no relevance to how many may be allowed to attend.

**Rationale3:** Rain would preclude the use of the stadium which has no limit of the number of family members attending and force the use of the gymnasium, which limits the number attending to three.

**Answer:** The answer is Rationale3

[...]

*Exemplars with Task Instruction*

**Context:** Because it permits a slower and more natural rhythm of life, living in the country is supposed to be more healthy and relaxed than living in the city. But surveys show that people living in the country become ill as often and as seriously as people living in the city, and that they experience an equal amount of stress.

**Question:** The statements above, if true, provide the most support for which one of the following?

**Rationale0:** This passage kind of disputes this line of thinking, living in the country should have a slower rhythm yet they experience the same amount of stress as a city dweller.

**Rationale1:** This passage is not saying this specifically, just that a natural rhythm might not have as many benefits as people want to believe.

**Rationale2:** The passage kind of focuses on both the thoughts that living in the country should be healthier but surveys show that it is not the case.

**Rationale3:** None of the above choices

**Answer:**

*Test Instance*

Figure 12: Example of the prompt used for rationale alignment task.

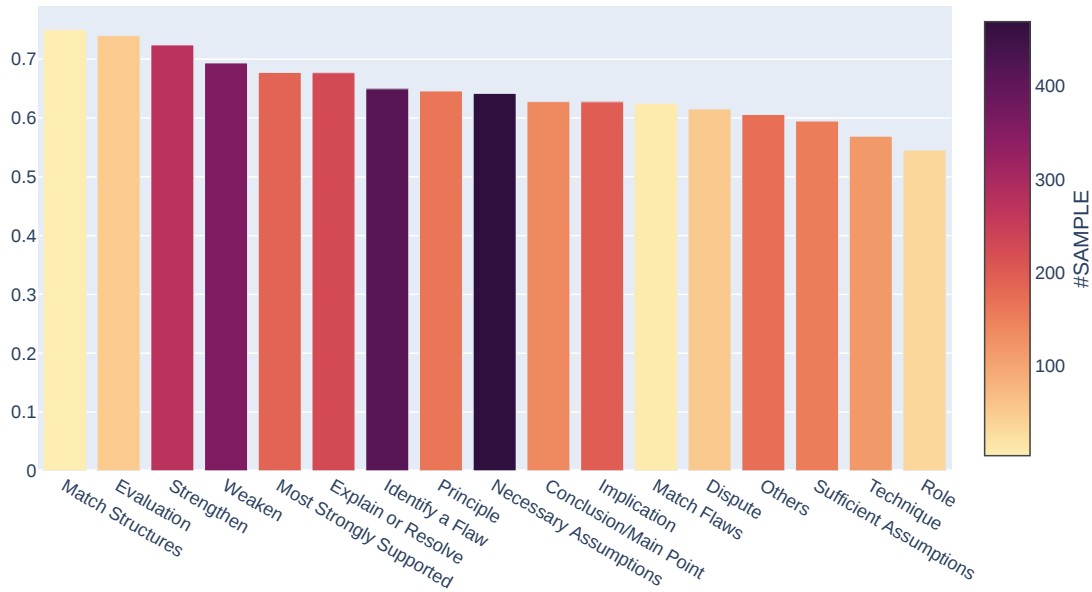

Figure 13: Accuracy of InstructGPT and reasoning types originally annotated in the ReClor dataset.

| Reasoning Type | Passage | Question | Option | Correct | Rationale |
|---|---|---|---|---|---|
| *Direct Contextual* | Trisha: Today's family is declining in its ability to carry out [...]. There must be a return to the traditional values of commitment and responsibility. Jerod: We ought to leave what is good enough alone. Contemporary families may be less stable than traditionally, but most people do not find that to be bad. [...]. | Trisha and Jerod disagree over whether the institution of the family is | no longer traditional. | FALSE | Both Trisha and Jerod agree that families are no longer traditional, this is not what the argument is about. |
| *Direct External* | A just government never restricts the right of its citizens to act upon their desires except when their acting upon their desires is a direct threat to the health or property of other of its citizens. | Which one of the following judgments most closely conforms to the principle cited above? | A just government would not censor writings of Shakespeare, but it could censor magazines and movies that criticize the government. | FALSE | A just government would not censor magazines and movies that criticize the government because these things do not threaten the health or property of its citizens. |
| *Indirect Contextual* | Doctor: The practice of using this therapy to treat the illness cannot be adequately supported by the claim that any therapy for treating the illness is more effective than no therapy at all. What must also be taken into account is that this therapy is expensive and complicated. | Which one of the following most accurately expresses the main point of the doctor's argument? | The therapy's possible effectiveness in treating the illness is not sufficient justification for using it. | TRUE | Therapy's other costs must be considered before enlisting the treatment as it is not cheap and not simple. |
| *Indirect External* | On average, corporations that encourage frequent social events in the workplace show higher profits than those that rarely do. This suggests that the EZ Corporation could boost its profits by having more staff parties during business hours. | Which one of the following, if true, most weakens the argument above? | Frequent social events in a corporate workplace leave employees with less time to perform their assigned duties than they would otherwise have. | FALSE | Frequent social events in a corporate workplace can re-energize employees, like a lunch break does. |

Table 10: Examples of the reasoning types with a passage, a question, an option, the correctness of the option, and its human-written rationale.

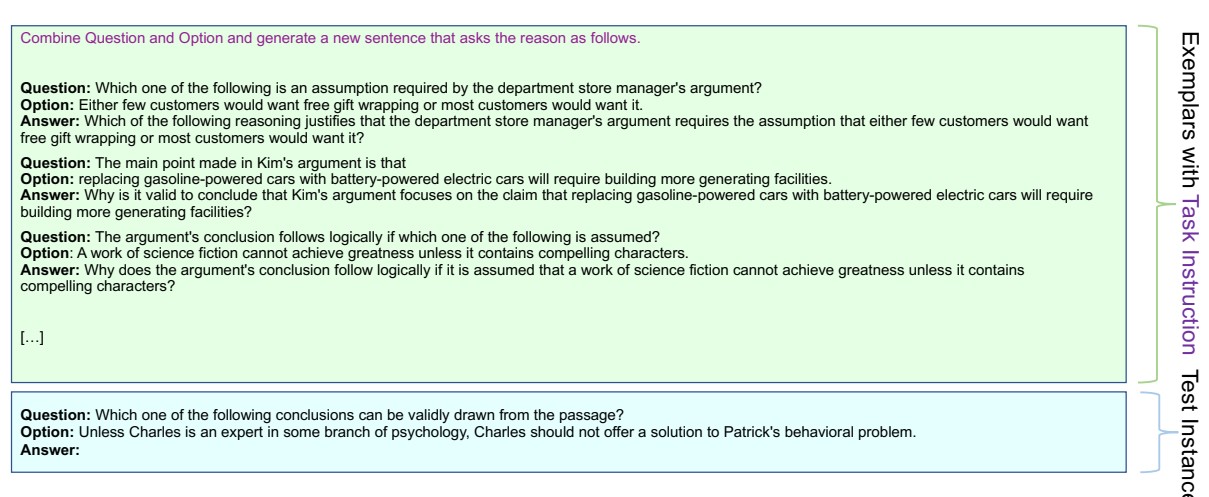

Figure 14: Example of the prompt used to generate subquestions.

| Paragraph: Trisha: Today's family is declining in its ability to carry out its functions of child-rearing and providing stability for adult life. There must be a return to the traditional values of commitment and responsibility. Jerod: We ought to leave what is good enough alone. Contemporary families may be less stable than traditionally, but most people do not find that to be bad. Contemporary criticisms of the family are overblown and destructive. |
|:---|

| | |
|:---|:---|
| MainQ | **Question:** Trisha and Jerod disagree over whether the institution of the family is
**Options:**
1) valued by most people.
2) changing over time.
**3**) adequate as it is.
4) no longer traditional. |
| Selective SubQ | **Question:** What is the source of the disagreement between Trisha and Jerod regarding whether the institution of the family is adequate as it is?
**Options:**
1) The argument does not mention value to the people.
**2**) Trisha is arguing that things were better with traditional families and Jerod is arguing that they are good now, the argument is about the quality of the relationship now.
3) Both Trisha and Jerod agree that families are no longer traditional, this is not what the argument is about.
4) None of the above choices. |
| Eliminative SubQ | **Question:** What evidence is there to suggest that Trisha and Jerod's disagreement over whether the institution of the family is no longer traditional is not valid?
**Options:**
**1**) Both Trisha and Jerod agree that families are no longer traditional, this is not what the argument is about.
2) Trisha is arguing that things were better with traditional families and Jerod is arguing that they are good now, the argument is about the quality of the relationship now.
3) The argument does not mention value to the people.
4) None of the above choices. |

Table 11: Examples of the main questions and subquestions in our dataset. The options in bold indicate the correct answer.

| Question Type | Option | Rationale |
|:---|:---|:---|
| Selective | Delays in the communication of discoveries will have a chilling effect on scientific research. | Delays in communicating discoveries would limit the time other scientists have to investigate and contribute. |
| | Kimmy is a highly compensated and extremely popular television and movie actress. | All the information in the passage indicates that Kimmy is affluent and renowned. |
| | Before new therapeutic agents reach the marketplace, they do not benefit patients. | The passage states that new therapies aid patients only after they are introduced to the marketplace. |
| Eliminative | The speed of eye orientation correlates with intelligence, not overall health. | The speed at which one can orient one's eye to a stimulus has been closely associated with overall health. |

Table 12: Rationales that are semantically similar to the MainQ's option in Selective and Eliminative SubQs.

*Exemplars with CoT*

*Test Instance*

Figure 15: Example of the prompt using the chain-of-thought approach.

We are collecting data to assess how well machines understand questions that require complex logical reasoning such as in GMAT or LSAT. We appreciate your help!

In this task, you are given a passage, question about it, and four answer options where one option is the correct answer and the others are incorrect (the check and cross marks indicate it). For each option, you are asked to write **rationale that is implicitly assumed by the question writer and supports why the option are correct or incorrect.**

We recommend spending 5-10 minutes on one HIT.

### Guidelines for Writing Rationales

- Write the rationale in one sentence.
- Not only extract a fact from the passage, but also write **the reasoning process and background knowledge** that are not written in the passage explicitly, in your own words.
- Write **concrete** facts and knowledge (See an example below). For example, make sure to include proper nouns that appear in the passage and option you focus on.
- You have to write rationales as **self-contained**: You shouldn't use pronouns that refer to expressions in the main question and options when writing rationales.
- Do not include any text that explicitly states that the option is correct or incorrect.(e.g., "This options is correct one because~", "~, so this option is wrong.")
- **UPDATED: When considering the rationale for a question involving "if true X," focus on the role that X plays in the passage under the assumption that it is true, rather than whether X is actually true.**

**To ensure the quality of rationale, please note the following two points.**

- Particularity
  Avoid writing **generic facts that are applicable to the other options** (e.g., "this option is incorrect because it doesn't match described events in the passage")
  In other words, for each option, write a rationale that **only applies to that option.**
  This is further rephrased to say that you should write rationales so that others should be able to associate each rationale with the choice.
  **UPDATED: When writing an rationale, try incorporating some of the content of the corresponding option and explain how it relates to the passage and the question.**
- Essentiality
  Write a rationale that is essentially necessary to justify whether the option is correct or not. This means that the option changes its correctness if the given rationale is not valid.

Figure 16: Instructions for the rationale writing task (1/4).

**Example**

When Alicia Green borrowed a neighbor' s car without permission, the police merely gave her a warning. However, when Peter Foster did the same thing, he was charged with automobile theft. Peter came to the attention of the police because the car he was driving was hit by a speeding taxi. Alicia was stopped because the car she was driving had defective tail lights. It is true that the car Peter took got damaged and the car Alicia took did not, but since it was the taxi that caused the damage this difference was not due to any difference in the blameworthiness of their behavior. Therefore Alicia should also have been charged with automobile theft.

Question:

The statement that the car Peter took got damaged and the car Alicia took did not plays which one of the following roles in the argument?

Answer Options

- ✅ A: It demonstrates awareness of a fact on which a possible objection might be based.
- ❌ B: It illustrates a general principle on which the argument relies.
- ❌ C: It presents a reason that directly supports the conclusion.
- ❌ D: It justifies the difference in the actual outcome in the two cases.

In this example you have to write rationales that explain

- why A is correct answer.
- why B is not correct answer.
- why C is not correct answer.
- why D is not correct answer.

For example you can write rationales with regard to this question such as:

**Good Rationales**

A: The author assumes as a possible objection that the car damage caused the difference and denies it.
B: The statement that the car Peter took got damaged and the car Alicia took did not describes a fact which occurred in the passage, not a principle.
C: The difference in the damage to the car does not support the conclusion but is supposed to be the rationale for the opposite conclusion instead.
D: The damage to the car should not justify differences in actual outcomes because whether the taken car got damaged was just the reason why his theft was brought to light.

**Bad Rationales**

A: The statement is a fact on which a possible objection might be based.
B: The statement is not a general principle.
C: The statement doesn't directly support the conclusion.
D: The statement doesn't justify the difference in the actual outcome in the two cases.

**Reason**

The bad rationales are (1) ambiguous and (2) they refer to expressions used in main question(e.g., "The statement")

Write concrete rationales that well explain the reasoning and background knowledge for answering the question.

The following two examples focus on essentiality and particularity.

Figure 17: Instructions for the rationale writing task (2/4).

**Example(Essentiality)**

Black Americans are twice as likely to suffer from hypertension as white Americans.The same is true when comparing Westernized black Africans to white Africans.The researchers hypothesized that the reason why westernized black people suffer from hypertension is the result of the interaction of two reasons? one is the high salt content of western foods, and the other is the adaptation mechanism of black genetic genes to the salt-deficient environment.

Question

The following conclusions about contemporary westernized African blacks, if the item is true, can it best support the researcher's hypothesis?

Answer Option

❌ The blood pressure of Yoruba people in West Africa is not high. Yoruba people have lived inland far away from sea salt and far away from the Sahara salt mine in Africa.

**Bad Rationale**

Yoruba people are not Westernized.

**Reason**

This rationale is not an essential one.

That is because whether the Yoruba were westernized or not is not mentioned in the article, and most importantly, it is not relevant for determining the correctness of the option.

Even if it were true that the Yoruba were westernized, the fact that the Yoruba did not have higher blood pressure in salt-deficient environments contradicts the second reason the researchers gave, making that option incorrect.

A more appropriate rationale would be "the fact that the Yoruba do not have higher blood pressure in salt-deficient environments contradicts the researchers' hypothesis that they are more likely to develop high blood pressure due to adaptation mechanisms to salt-deficient environments.

**Example(Particularity)**

The passage and question are the same as above.

Answer Option

✅ The blood pressure of the descendants of Senegalese and Gambians is usually not high, and the history of Senegal and Gambia has not been short of salt.

**Bad Rationale**

Senegal and Gambia are African nations that are relevant to the study.

**Reason**

This rationale is too generic.

Instead, please describe the rationale in detail as follows, for example.

"The fact that the descendants of Africans who are not salt-deficient do not have high blood pressure suggests that high blood pressure is related to a chronic lack of salt."

Figure 18: Instructions for the rationale writing task (3/4).

For many people, leisure time is an opportunity to get outdoors, have some fun and meet interesting people. Add two pieces of advanced 21stcentury technology -- global positioning system (GPS) devices and the Internet -- to get ""geocaching"". The word geocaching comes from ""geo"" (earth) and ""cache"" (hidden storage). Geocachers log onto a website to find information about the location of a cache -- usually a waterproof plastic box containing small items such as toys and CDs -- along with a notebook where ""finders"" can enter comments and learn about the cache ""owner"", the person who created and hid the cache. Finders may take any of the items in the cache but are expected to replace them with something of similar value. They then visit the website again and write a message to the owner.

Geocaching became possible on May 1, 2000, when a satellite system developed by the U.S. Department of Defense was made public. Using an inexpensive GPS device, anyone on earth can send a signal to the satellites and receive information about their position. This is basically a high-tech version of orienteering, the traditional pastime which uses maps and compasses instead of GPS to determine one's location.

Geocachers are a very considerate group. Owners carefully choose a cache's location to give finders an enjoyable experience, such as a beautiful view or a good campsite. They also consider the environmental impact of their cache since it could result in an increased number of visitors to an area. As for the content of the caches, owners and finders must only use items that are suitable for the whole family, as caches are found by geocachers of all ages

Question

Which of the following is true according to the passage?

Answer Options

    ❌ A: Most geocachers are adults.
    ❌ B: Any item can be placed in the caches.
    ❌ C: The caches should be put in a remote place.
    ✅ D: Geocachers try to avoid damaging the environment.

**Rationales**

    A: All age groups participate in geocaching.
    B: The finder is expected to put into the cache something as valuable as the items in the cache.
    C: Geocachers consider the placement of the cache based mainly on the fun for the finder.
    D: Geocachers also consider the consequences of increasing the number of visitors due to the popularity of geocaching.

If we want to deal with the association between boys and girls properly, here are some "dos and don'ts" for you to follow. Keep a normal and healthy state of mind. Our schools and classes are made up of boys and girls. It is very natural for boys and girls to make friends with each other. We should make as many friends as possible. We should keep in touch with the other sex in public instead of in secret.

Don't be too nervous or too shy. If you are a shy person, you can also find a way out. First of all, you can make friends with the students who have the same interest and hobby as you. As both of you have much in common, you may have much to talk about. If you keep doing like that, little by little, you will gladly find you are also as free to express yourself as others. Don't fall into the trap of early love. The boys and girls at adolescence are rich in feeling. They are easy to regard the friendship as a sign of love and fall in love with each other at an early age. In my opinion, early love is a green apple that can't be eaten. An apple won't taste sweet until it is fully ripe. Boys and girls at middle school are too young to carry the heavy duty of love. Do keep out of early love.

Question

The main idea of the passage is to _ .

Answer Options

    ❌ A: tell students to keep away from early love.
    ✅ B: give some advice on how to associate between boys and girls.
    ❌ C: tell students how to make friends.
    ❌ D: teach boys how to talk with girls.

**Rationales**

    A: The author recommend staying away from early love as part of encouraging healthy relationships between men and women.
    B: The author offers several notes on how to maintain healthy relationships between men and women.
    C: Making friends is juts a part of building proper relationships.
    D: The author tells general ways how to talk with others.

Figure 19: Instructions for the rationale writing task (4/4).

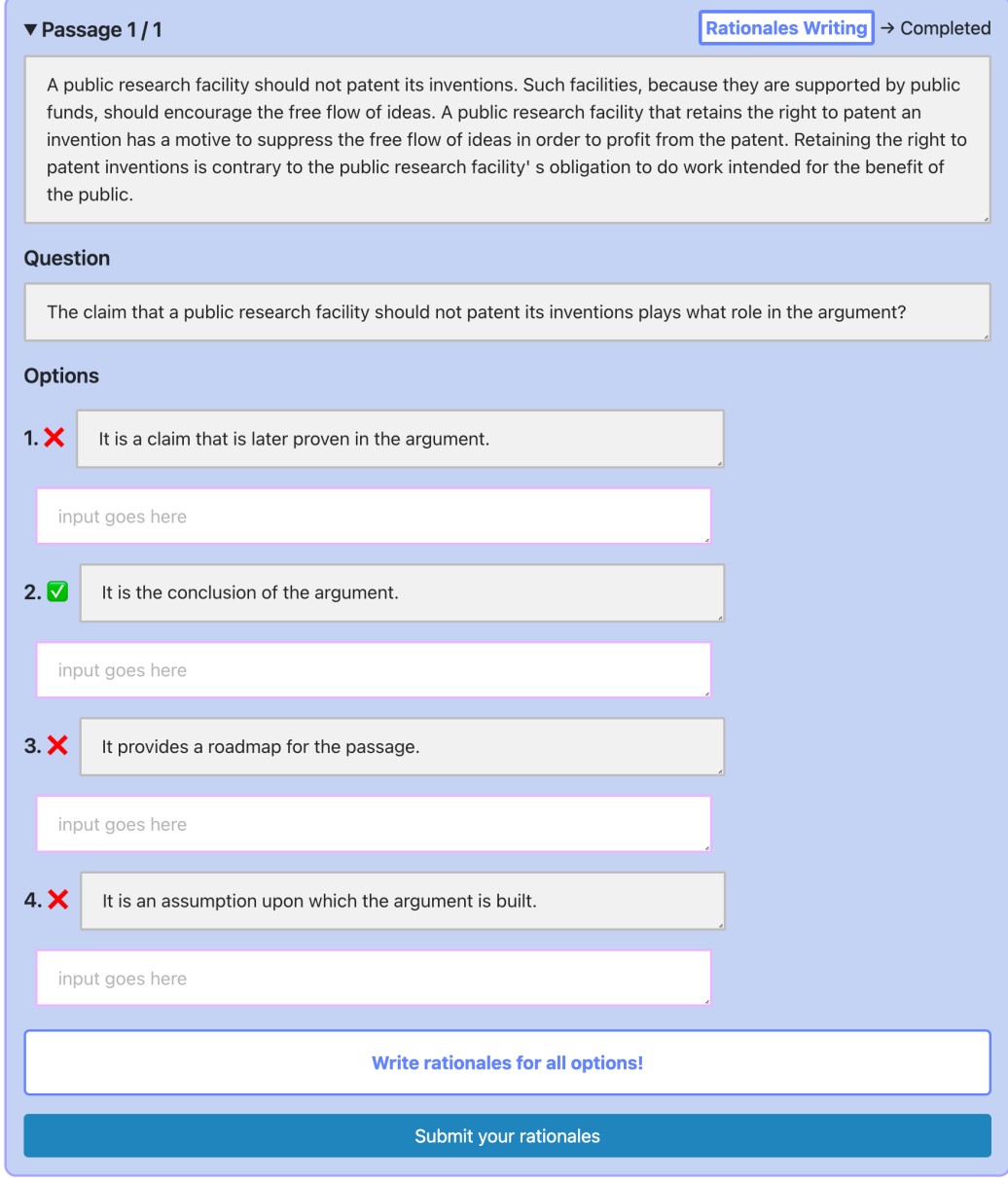

This is a task of data collection for studying machines' reading comprehension technology. Given the passage, question, and four answer options, you are asked to write **rationales explaining why each option is correct or incorrect**.
The check and cross marks indicate the correctness of the options. Please read the full instruction before starting. (We have **updates** in red texts.)

Notes
- Do not include any text that explicitly states that the option is correct or incorrect. (e.g., "this option is incorrect because it doesn't match described events in the passage")
- To ensure the quality of rationale, please note the following two points.
  - Particularity
    Avoid writing **generic facts that are applicable to the other options** (e.g., "this option is incorrect because it doesn't match described events in the passage")
    In other words, for each option, write a rationale that **only applies to that option.**
    This is further rephrased to say that you should write rationales so that others should be able to associate each rationale with the choice.
  - Essentiality
    Write a rationale that is essentially necessary to justify whether the option is correct or not. This means that the option changes its correctness if the given rationale is not valid.

**You can accept up to 20 HITs for this batch.**

▼ Passage 1 / 1     [Rationales Writing] → Completed

A public research facility should not patent its inventions. Such facilities, because they are supported by public funds, should encourage the free flow of ideas. A public research facility that retains the right to patent an invention has a motive to suppress the free flow of ideas in order to profit from the patent. Retaining the right to patent inventions is contrary to the public research facility' s obligation to do work intended for the benefit of the public.

**Question**

The claim that a public research facility should not patent its inventions plays what role in the argument?

**Options**

1. ❌   It is a claim that is later proven in the argument.

    input goes here

2. ✅   It is the conclusion of the argument.

    input goes here

3. ❌   It provides a roadmap for the passage.

    input goes here

4. ❌   It is an assumption upon which the argument is built.

    input goes here

**Write rationales for all options!**

Submit your rationales

Figure 20: Rationale writing task interface. The workers are given a context, question, and four options along with their correctness, and are asked to provide a rationale for each choice.

We are assessing the quality of rationale written for multipe-choice reading comprehension questions.

In this task, you are given a passage, question about it, and four answer options where one option is the correct answer and the others are incorrect (the check and cross marks indicate it).

In addition, you are given a rationale, which corresponds to one of the options and its correctness.

You are asked to **answer which option the given rationale corresponds to** by checking the radio button.

We recommend spending about 1-2 minutes on one HIT.

**Example**

> The town council of North Tarrytown favored changing the name of the town to Sleepy Hollow. Council members argued that making the town' s association with Washington Irving and his famous "legend" more obvious would increase tourism and result immediately in financial benefits for the town' s inhabitants.

Question:

The council members' argument requires the assumption that

Answer Options

✅ A: the immediate per capita cost to inhabitants of changing the name of the town would be less than the immediate per capita revenue they would receive from the change.

❌ B: other towns in the region have changed their names to reflect historical associations and have, as a result, experienced a rise in tourism.

❌ C: the town can accomplish, at a very low cost per capita, the improvements in tourist facilities that an increase in tourism would require.

❌ D: many inhabitants would be ready to supply tourists with information about Washington Irving and his "legend".

Rationale

Creating tourist facilities is not necessary to benefit from the financial gains of changing the towns name.

In this example, the answer is C. That's because the rationale explains why optionC is not correct, i.e., that building traveler's facility is not assumed in the passagee.

Figure 21: Instructions for the rationale validation task.

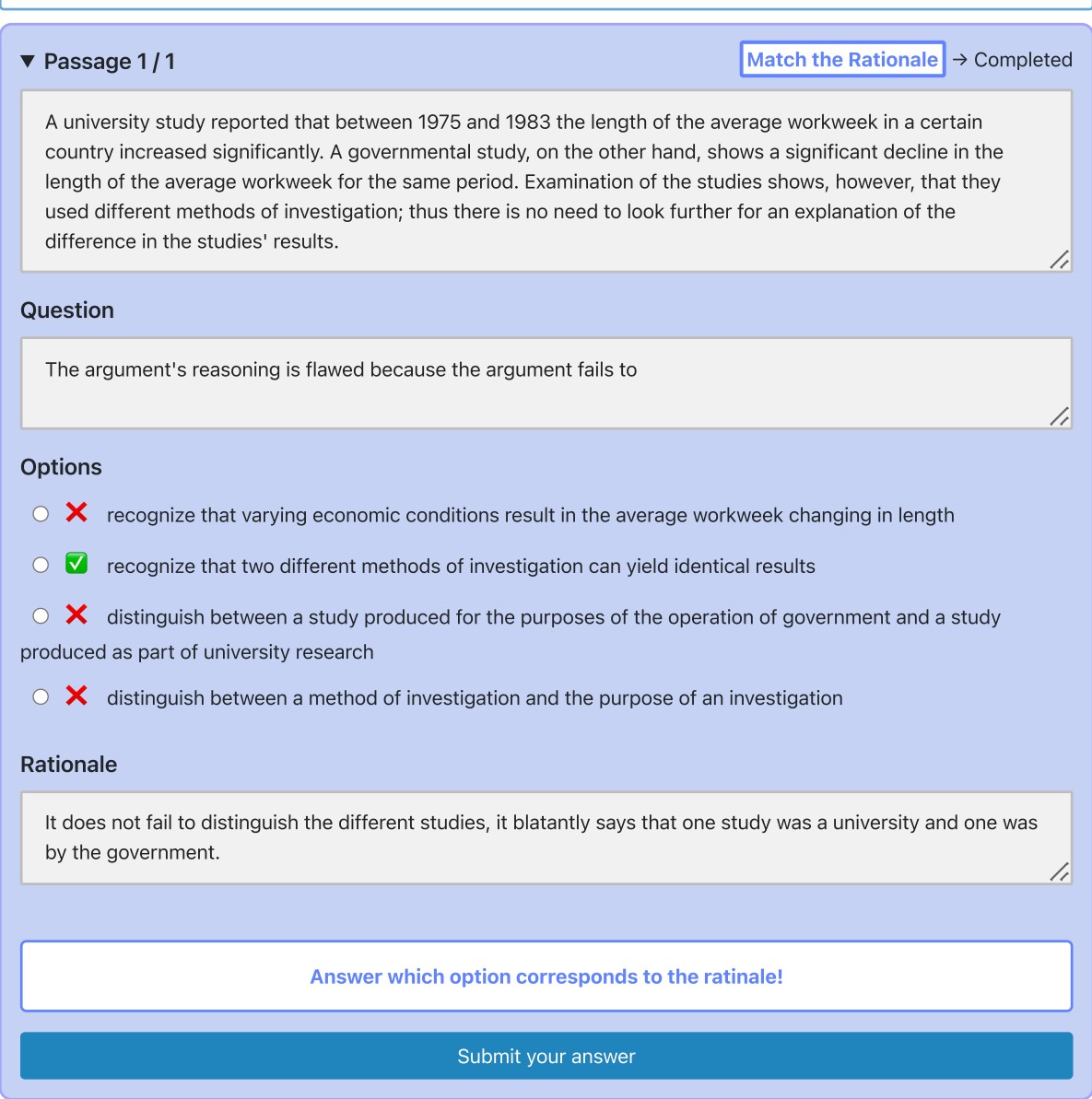

# Evaluate explanation for reading comprehension

**Instructions**

In this task, you are given a passage, question about it, and four answer options where one option is the correct answer and the others are incorrect (the check and cross marks indicate it).
In addition, you are given a rationale, which corresponds to one of the options and its correctness.
You are asked to **answer which option the given rationale corresponds to** by checking the radio button.
You can accept up to 160 HITs for this batch.

▼ **Passage 1 / 1**                    **Match the Rationale** → Completed

A university study reported that between 1975 and 1983 the length of the average workweek in a certain country increased significantly. A governmental study, on the other hand, shows a significant decline in the length of the average workweek for the same period. Examination of the studies shows, however, that they used different methods of investigation; thus there is no need to look further for an explanation of the difference in the studies' results.

**Question**

The argument's reasoning is flawed because the argument fails to

**Options**

○ ✗   recognize that varying economic conditions result in the average workweek changing in length

○ ✅   recognize that two different methods of investigation can yield identical results

○ ✗   distinguish between a study produced for the purposes of the operation of government and a study produced as part of university research

○ ✗   distinguish between a method of investigation and the purpose of an investigation

**Rationale**

It does not fail to distinguish the different studies, it blatantly says that one study was a university and one was by the government.

**Answer which option corresponds to the ratinale!**

**Submit your answer**

Figure 22: Rationale validation task interface. The workers select the option best supported by the provided rationale.

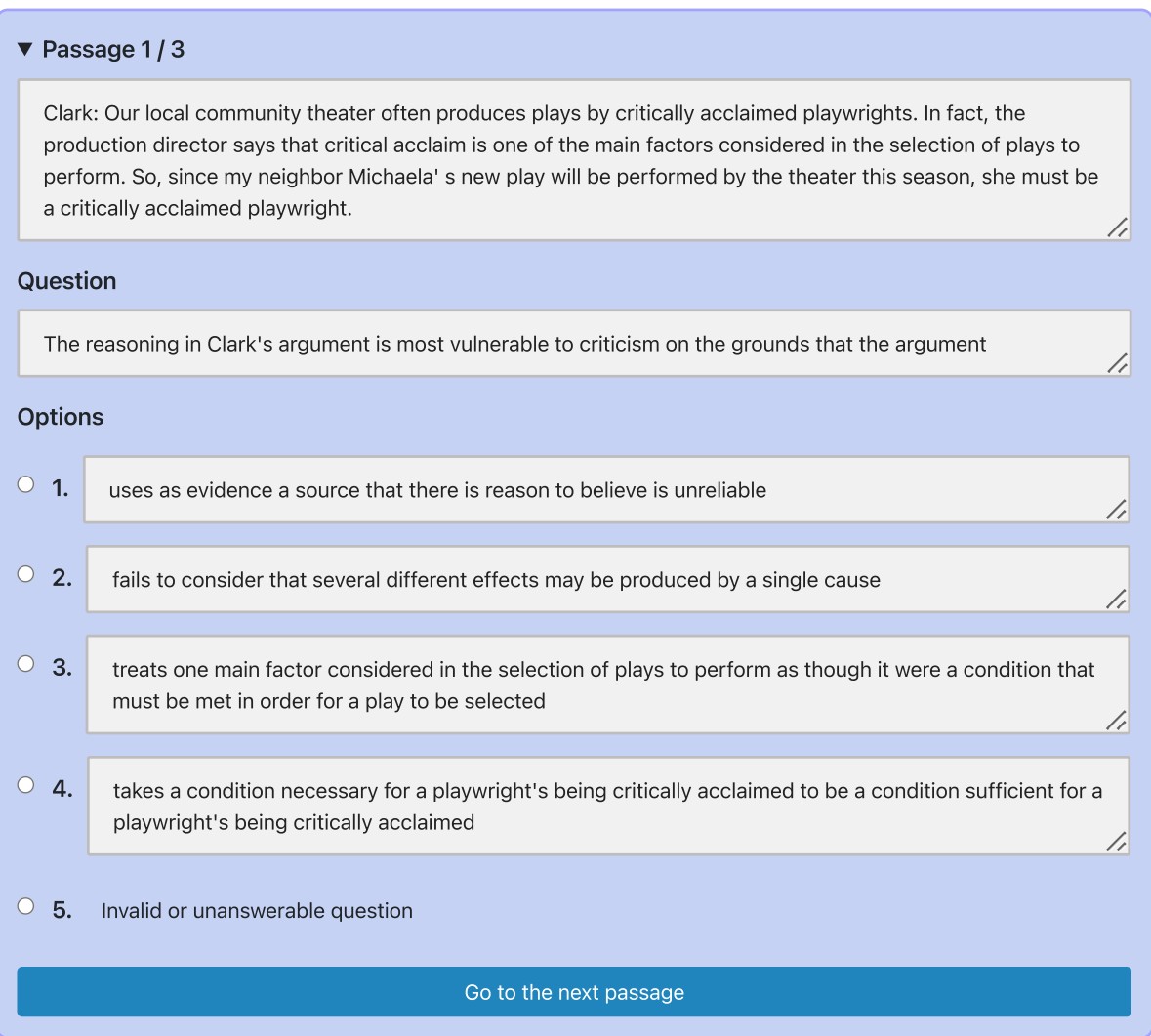

Figure 23: Human validation task interface. The workers are asked to answer subquestions.