# OpenReview forum: "Evaluating the Rationale Understanding of Critical Reasoning in Logical Reading Comprehension"
_EMNLP/2023/Conference — EMNLP 2023 Main_

### Official Review · Reviewer_X3Ra · 2023-07-24

**Paper Topic And Main Contributions:** 1. The authors introduce the RULE dat…
**Typos Grammar Style And Presentation Improvements:** 1. Broken citation on line 040 (“; ?”…
**Soundness:** 4

**Excitement:**

4: Strong: This paper deepens the understanding of some phenomenon or lowers the barriers to an existing research direction.

**Questions For The Authors:**

1. I found the sub-question provided in Figure 1 a little difficult to parse. Did you try alternative ways of phrasing the sub-questions?
For example:
Why is “deriving implications of a generalization that it assumes to be true” not the correct answer for the question “The argument proceeds by doing which one of the following”?
(I think this phrasing might be easier for both humans and machines to parse)

**Reasons To Accept:**

1. I think that the RULE dataset would be useful for future research, particularly for improving language models’ reasoning abilities
2. The examination of language model limitations show areas where language models can potentially be improved

**Reasons To Reject:**

1. The RULE dataset is built on the ReClor dataset. However, ReClor is not released under an open source/data license. It is unclear if the authors obtained permission to use and redistribute this data. Licensing information is also not provided for the RULE dataset, so it is unclear how and if this new dataset can be used by other researchers.

Edit: The authors have clarified in there rebuttal that they have obtained a license for ReClor. However, it is still unclear if the RULE dataset will be made available and how it will be licensed

Edit 2: The RULE dataset will be made publicly available under the CC BY-NC 4.0 license

**Reproducibility:**

3: Could reproduce the results with some difficulty. The settings of parameters are underspecified or subjectively determined; the training/evaluation data are not widely available.

**Reviewer Confidence:**

4: Quite sure. I tried to check the important points carefully. It's unlikely, though conceivable, that I missed something that should affect my ratings.

---

> ### Author Rebuttal · Authors · 2023-08-27
>
> We would like to express our gratitude for the time and effort you have dedicated to reviewing our manuscript.
>
>
> >License of ReClor:
>
> We reached out to the authors of ReClor via email to inquire about the licensing for using ReClor as a part of our dataset. In response, we received confirmation that a portion of ReClor is permitted under the use item of ReClor, which can be confirmed on their project page. We will make sure to clarify this information in our revised manuscript.
>
>
> >Alternative Ways of Phrasing the Sub-questions:
>
> In addition to generating from the main question (Q) and the options (Op) using InstructGPT, we also attempted to transform Q+Op into declarative sentences and subsequently into interrogative ones. Although we used T5 and QA2D (Demszky etal., 2018) for the transformation into declarative sentences, neither method generated high-quality declarative sentences.
>
> We appreciate your suggested heuristic approach to constructing subquestions, emphasizing ease of parsing. While we understand and acknowledge the advantages of your proposed phrasing, our main intent was to ensure that the subquestions are self-contained (i.e., excluding phrases like "which of the following" from Q that necessitate providing the original options). In hindsight, a generation method that modifies the Q part of your heuristic to be self-contained may have been a better approach. For example: Why is “deriving implications of a generalization that it assumes to be true” not the correct answer for the question "What method or technique does the argument employ?"
>
>
> Thank you once again for your valuable feedback.
>
> Reference
> Demszky, D., Guu, K., & Liang, P. (2018). Transforming Question Answering Datasets Into Natural Language Inference Datasets. arXiv preprint. arXiv:1809.02922.

---

### Official Review · Reviewer_i432 · 2023-08-05

**Soundness:** 4

**Excitement:**

4: Strong: This paper deepens the understanding of some phenomenon or lowers the barriers to an existing research direction.

**Missing References:**

(tangentially relevant): https://arxiv.org/abs/2204.05212 and https://arxiv.org/abs/2210.10860v1 also collect explanations for each answer choice, albeit for a different purpose. Unlike this work, their explanations for wrong answer argue for why answer should be accepted as opposed to eliminated.

L040: missing reference


**Paper Topic And Main Contributions:**

This work evaluates models' logical reasoning in the MCQ setting by probing it with additional questions about the reasoning behind selecting or eliminating individual choices. To this end, the authors build a new dataset, RULE. They start with questions from the ReClor dataset and annotate rationales for selecting and eliminating each of its answer choices. They then generate questions like for each choice which has one of the generated rationales as the answer.

These additional (sub-)questions have 2 key features:

1. They are in the same MCQ format as the original question. So one can probe the models' reasoning on the original question by these additional questions.
2. They are contrastive (minimally different), i.e., all subquestions share the same passage and answer choices, but they have different answers depending on the subquestion. This feature prevents models from taking simple shortcuts.

The authors have ensured the annotations are of high quality by human validation, and have also established a very high human score on the task.

Finally, the authors benchmark a large number of recent (few-shot, fine-tuned, with/without CoT) models on ReClor and RULE and demonstrate that all the models struggle and are behind humans. In particular, they find that models are extremely bad at selecting why a given choice is incorrect.

Finally, they explore (i) model-generated rationales and find humans are better at this task. (ii) using human-written rationales to improve models and find that selective subquestions help, eliminative ones hurt.

**Questions For The Authors:**

Suggestion:

A. I have a different hypothesis about why the score for eliminative subquestions is so much worse than selective subquestions: It is because the models ignore the word "not". It would be interesting to do an experiment to test this: Compare model predictions for eliminative subquestions with and without the word "not" and see if the scores are close and individual responses correlate. Although not necessary, it'd be a good addition to the paper (at least the appendix, if not the main paper).

**Reasons To Accept:**

- This is a very well-done study and evaluation design. I particularly liked the two features highlighted in the summary above.
- This is a nice high-quality resource, and should be helpful for people exploring logical reasoning models.
- I was impressed by the number and diversity of the models that they have benchmarked for this new task (both fine-tuned and few-shot).

**Reasons To Reject:**

I don't see any reason to reject this work.


**Reproducibility:**

5: Could easily reproduce the results.

**Reviewer Confidence:**

4: Quite sure. I tried to check the important points carefully. It's unlikely, though conceivable, that I missed something that should affect my ratings.

---

> ### Author Rebuttal · Authors · 2023-08-27
>
> Firstly, we would like to express our gratitude for pointing out the fascinating hypothesis and  the related literature. Your comments are valuable to our work.
>
>
> >Comparison of Model Predictions for Eliminative Subquestions with and without the Word "not"
>
> We investigated the hypothesis under the following settings.:
> - Extracted eliminative subquestions that include the word "not" (n = 2156).
> - Compared the performance of text-davinci-003 by providing 5-shot exemplars from ReClor as a prompt on the subset (vanilla) and on the subset where "not" is eliminated from the question (elim_not).
>
>
> **Results**
>
> Accuracy:
> - Vanilla: 57.33±1.61%
> - Elim_not: 36.26±4.77%
>
> Cohen's Kappa: 0.207
>
> 2*2 Matrix:
>
> |          | Vanilla Correct | Vanilla Incorrect |
> |----------|----------|----------|
> | **Elim_not Correct** | 566  | 670  |
> | **Elim_not Incorrect** | 216  | 704  |
>
> From the results above, we conclude that the presence of "not" significantly affects the model's reasoning. However, it is interesting that the model was still capable of answering 36% of the eliminative subquestions even when the polarity of the question was reversed (i.e., after the negation was removed).
> We will add these results in our revised manuscript.
>
> We deeply appreciate your insightful comments, which have improved our work.
> Thank you for taking the time to review our manuscript.

---

### Official Review · Reviewer_ivgz · 2023-08-06

**Soundness:** 4

**Excitement:**

4: Strong: This paper deepens the understanding of some phenomenon or lowers the barriers to an existing research direction.

**Paper Topic And Main Contributions:**

This paper proposes a dataset composed of auxiliary questions and rationales to test the model’s consistent ability for critical reasoning. The paper presents experiments that compare various models with different conditions on this task. Additional analysis on difficulties of eliminative subquestions and comparison between rationale writing ability also provides insights into the model’s behavior on reasoning tasks.

**Questions For The Authors:**

Question A. (Line 70-72) What are the examples of selection elimination process of relevant alternatives in logical reasoning?

Question B. (Line 182-184) Why is faithfully testing the model’s performance on the main questions important?



**Reasons To Accept:**

1. The problem is well-motivated by grounding on the prior work.
2. The paper contributes valuable human-written free-from rationales datasets. The choice of the dataset is well-motivated and details about the annotation and qualification process are thoroughly performed. Especially, qualification and essential criteria for writing rationales are concrete. (It would be good to have examples of comparison between specific and unspecific samples and consistent and inconsistent samples)
3. The experiment setting is concrete and detailed enough to reproduce and findings are well-organized in that there are also comparisons between findings from prior works.


**Reasons To Reject:**

Annotation of rationales: Annotators can write the rationales that are the semantically same with the option but with different expressions. Can this kind of rationale be filtered out with current verification? Or is this kind of rationale added to the dataset?


**Reproducibility:**

4: Could mostly reproduce the results, but there may be some variation because of sample variance or minor variations in their interpretation of the protocol or method.

**Reviewer Confidence:**

4: Quite sure. I tried to check the important points carefully. It's unlikely, though conceivable, that I missed something that should affect my ratings.

**Typos Grammar Style And Presentation Improvements:**

Line 40: reference info is missing
Line 252-264: Explaining necessity with consistency is confusing

---

> ### Author Rebuttal · Authors · 2023-08-27
>
> We sincerely appreciate the invaluable feedback and constructive comments from the reviewer.
>
>
> >On the Annotation of Rationales: Annotators can write the rationales that are the semantically same with the option but with different expressions.
>
> We did not have a filtering for such rationales. To confirm the degree of the similarity, we performed a manual annotation of our dataset, extracting 50 pairs of main options and rationales, both for selective and eliminative rationales.
>
> For selective rationales, we identified three out of 50 pairs as semantically similar:
> 1. Main Option: “Delays in the communication of discoveries will have a chilling effect on scientific research.”
> Rationale: “Delays in communicating discoveries would limit the time other scientists have to investigate and contribute.”
> 2. Main Option: “Kimmy is a highly compensated and extremely popular television and movie actress.”
> Rationale: “All the information in the passage indicates that Kimmy is affluent and renowned.”
> 3.
> Main Option: “Before new therapeutic agents reach the marketplace, they do not benefit patients.”
> Rationale: “The passage states that new therapies aid patients only after they are introduced to the marketplace.”
>
> For eliminative rationales, we noted only one similar pair:
> 1. Main Option: “The speed of eye orientation correlates with intelligence, not overall health.”
> Rationale: “The speed at which one can orient one's eye to a stimulus has been closely associated with overall health.”
>
>
> >The examples of selection and elimination process of relevant alternatives in logical reasoning (Question A):
>
> The process of selecting or eliminating relevant alternatives in logical reasoning involves discerning incorrect possibilities and choosing the correct one.
> In the context of multiple-choice questions, this process refers to ruling out incorrect options to identify the right option. Our dataset evaluates this capability by asking models to choose a rationale for each option.
> The reasoning involving such a process might be:
> "Option A is incorrect because this passage provides examples to support a claim, not to generalize it. [...] Option D is correct as the passage builds its conclusion by citing various sources."
>
>
> >Why is faithfully testing the model’s performance on the main questions important? (Question B):
>
> We apologize if our phrasing in L182-184 caused confusion. Our intent was to express that we aim to evaluate the model's understanding of the main questions through the auxiliary subquestions, ensuring a comprehensive evaluation. We will make this clearer in our revised manuscript.
>
>
> Thank you again for the insightful comments and the opportunity to clarify our work. We hope that our responses adequately address the concerns.

---

### Meta-Review · Area_Chair_WLpg · 2023-09-12

**Recommendation:** 5

**Metareview:**

All reviewers agree that the new RULE dataset on understanding rationales behind critical reasoning is a valuable resource, and that the paper is well motivated and well-structured.
The reviewers questions and concerns regarding licensing of the resource could be resolved during the discussion.

---

### Decision · Program_Chairs · 2023-10-07

**Decision:**

Accept-Main

**Comment:**

All reviewers agree that the new RULE dataset on understanding rationales behind critical reasoning is a valuable resource, and that the paper is well motivated and well-structured.
The reviewers questions and concerns regarding licensing of the resource could be resolved during the discussion.